# Human Expertise Really Matters! Mitigating Unfair Utility Induced by Heterogeneous Human Expertise in AI-assisted Decision-Making

## Abstract

AI-assisted decision-making often involves an AI model providing confidence, which helps human decision-makers integrate these with their own confidence to make higher-utility final decisions. However, when human decision-makers are heterogeneous in their expertise, existing AI-assisted decision-making may fail to provide fair utility across them. Such unfairness raises concerns about social welfare among diverse human decision-makers due to inequities in access to equally effective AI assistance, which may reduce their willingness and trust to engage with AI systems. In this work, we investigate how to calibrate AI confidence to provide fair utility for human decision-makers. We first demonstrate that rational decision-makers with heterogeneous expertise are unlikely to obtain fair decision utility from existing AI confidence calibrations. We propose a novel confidence calibration criterion, *inter-group-alignment*, which synergizes with human-alignment to jointly determine the upper bound of utility disparity across human decision-maker groups. Building on this foundation, we propose a new fairness-aware confidence calibration method, *group-level multicalibration*, which ensures a sufficient condition for achieving both inter-group-alignment and human-alignment. We validate our theoretical findings through extensive experiments on four real-world multimodal tasks. The results indicate that our calibrated AI confidence facilitates fairer utility, concurrently enhancing overall utility. *The implementation code is available at* https://anonymous.4open.science/r/iclr4103.

## 1 Introduction

In recent years, artificial intelligence (AI) has been increasingly leveraged to assist human decision-makers in decision-making across various domains. For example in typical binary classification tasks, AI systems have been developed to support clinicians in medical diagnosis (Rajpurkar et al., 2020; Wysocki et al., 2023), aid financial institutions in credit risk assessment (Bussmann et al., 2021), and assist legal professionals in bail or sentencing judgments (Dement & Inglis, 2024; Grgić-Hlača et al., 2019). However, AI is trained on datasets with inherent uncertainties and is still far from perfectly accurate in many real-world applications (Prabhudesai et al., 2023); **human decision-makers always need to integrate their own expertise with AI-generated insights to ensure the appropriateness and accuracy of final decisions.** One effective way to achieve this is by providing AI's confidence, which enables human decision-makers to better interpret the model's outputs (Bhatt et al., 2021; Steyvers & Kumar, 2024; Ma et al., 2023). Ideally, human decision-makers rely more on AI in situations where the AI's confidence is high and more on their own when the AI's confidence is low.

Existing research in AI-assisted decision-making has primarily focused on enhancing final decision-making's utility (the effectiveness of decisions, such as accuracy in classification tasks or prediction errors in regression tasks). Early studies suggested that AI confidence should be well-calibrated estimates of the probability that the predicted label matches the truth label (Pakdaman Naeini et al., 2015; Yin et al., 2019; Zhang et al., 2020). For instance, a well-calibrated diagnostic AI model expresses confidence $0.75$, to match the likelihood that the patient has the condition $P(Y = 1) = 0.75$. However, Vodrahalli et al. (2022b) experimentally demonstrated that in certain scenarios, explicitly uncalibrated AI advice led to substantially higher decision utility compared to well-calibrated advice

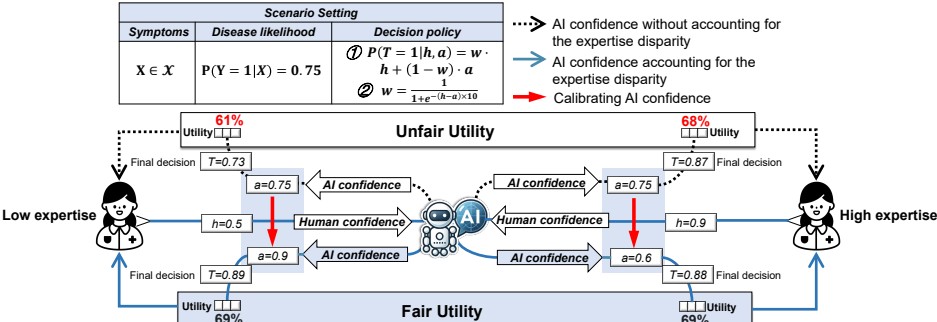

Figure 1: Illustration of AI-assisted decision-making where AI confidence is (1) calibrated without considering expertise disparities (**black dashed line**) and (2) calibrated considering expertise disparities (i.e., blue solid line). Our goal is to mitigate unfair utility across human decision-makers with heterogeneous expertise by calibrating AI confidence (i.e., red solid line). The utility is quantified by the accuracy $P(T = Y)$.

above. Subsequently, Corvelo Benz & Rodriguez (2023) provided a detailed theoretical analysis, demonstrating that rational decision-makers make optimal final decisions when AI confidence exhibits a natural alignment with human decision-makers' confidence in their own predictions, referred to as human-alignment.

Due to historical or unavoidable social factors, **human decision-makers may have varying expertise, making their confidence *not* align with the probability of truth labels in those observations. Existing AI confidence calibrations may result in unfair utility for human decision-makers with varying expertise.** Consider the AI-assisted medical diagnosis scenario illustrated in Figure 1. Human decision-makers are divided into two groups: low-expertise and high-expertise. Upon observing symptoms ($X$), they demonstrate different confidence in their diagnoses. High-expertise decision-makers have greater confidence ($h = 0.9$) in diagnosing a specific disease compared to those with low expertise ($h = 0.5$). Taking the case where human decision-makers follow a decision policy $P(T = 1|h, a) = w \cdot h + (1 - w) \cdot a$ as an example: if the AI provides an undifferentiated confidence ($a = 0.75$) without accounting for expertise disparities—it leads to unfair utilities: low-expertise decision-makers receive a utility of $0.61$, while high-expertise decision-makers receive a utility of $0.68$. **In this work, we aim to mitigate such utility disparity caused by heterogeneous expertise in AI-assisted decision-making.** A fairer utility provided by AI systems to human decision-makers with expertise disparity can increase human decision-makers willingness and trust to engage with AI assistance. In advancing the use of AI for good to improve social welfare, ensuring fair utility is particularly important. For example, in the case of the AI-assisted medical diagnosis above, ensuring fair AI support could help reduce the diagnostic error gap between less experienced doctors and experts by providing them with valuable decision-making experience. This, in turn, can help mitigate the *Matthew Effect* (Merton, 1968), which describes how disparities in resources such as education, economics, and information tend to widen, leaving the less advantaged further behind.

**Our contributions.** To the best of our knowledge, we present **the first work** that focuses on fairness issues arising from human decision-makers with heterogeneous expertise. ❶ Our first key contribution is the theoretical analysis showing that existing AI confidence mechanisms, including calibration and human-alignment, may not guarantee fair utility across human decision-makers with diverse expertise. ❷ The second key contribution is the critical concept of *inter-group-alignment*, which measures the disparity in the relationship between AI confidence and truth labels across different human decision-makers groups. This concept serves as a novel criterion for AI confidence calibration to ensure fair utility across human decision-makers groups with heterogeneous expertise. Additionally, we establish a tight upper bound of utility disparity in AI-assisted decision-making, determined by both levels of human-alignment and inter-group-alignment, offering insights into how AI confidence can be calibrated to help rational decision-makers achieve optimal and fair utility. ❸ To achieve the above calibration goals, we propose a new calibration approach *group-level multicalibration* inspired by multicalibration Hebert-Johnson et al. (2018), which is theoretically

proven to be a sufficient condition for simultaneously achieving both human-alignment and inter-group-alignment. ❹ To validate the practicality of our theoretical insights and the effectiveness of the proposed calibration method, we conduct extensive experiments on four AI-assisted decision-making tasks involving real human decision-makers. The results validate that calibrated AI confidence facilitates fair utility across diverse human decision-makers groups and enhances overall utility simultaneously.

## 2 PRELIMINARY

**Human-AI interactive model in AI-assisted decision making.** We focus on a binary decision-making scenario to investigate the existence of unfair utility in existing AI confidence calibrations and how to construct AI confidence values that ensure fair utility to human decision-makers groups with heterogeneous expertise. Binary decision-making is prevalent in real-world applications such as loan approvals, disease diagnoses, and job assignments. We illustrate the AI-assisted decision-making process in Figure 1. Factors that may lead to varying levels of expertise among human decision-makers include education level, job position, and personal characteristics such as gender, age, etc. We define the attribute used to group the human decision-makers as $S \in \mathcal{S}$. Let $f_H : \mathcal{X} \to [0, 1]$ represent the human decision-maker's confidence function regarding positive outcomes. Initially, the human decision-maker observes a sample with features $x \in \mathcal{X}$ and assigns a confidence $h = f_H(z) \in \mathcal{H}$. Subsequently, the AI model (i.e., a classifier) provides its confidence value $a = f_A(x, h, s)$ with $h$ and $s$ are optional variables, where $f_A : \mathcal{Z} \to [0, 1]$ denotes the AI's confidence function toward positive outcomes with $\mathcal{Z} = \{\mathcal{X}, \mathcal{H}, \mathcal{S}\}$. Finally, the human decision-maker makes a binary decision $T$ based on the probability $P(T = 1) = \pi(h, a) \in \{0, 1\}$: $T = 1$ if $P(T = 1) \geq 0.5$, and $T = 0$ otherwise, where $\pi \in \Pi(\mathcal{H}, \mathcal{A})$ denotes the decision-making policy. Upon making this decision, the decision-maker receives a utility $u(T, Y) \in \mathbb{R}$ under the truth label $Y \in \{0, 1\}$.

**Utility.** A natural setting for a utility function, consistent with most real-world scenarios, assigns higher utility to cases where the final decision, $T$, aligns with the ground truth label, $Y$, compared to cases where $T$ and $Y$ diverge. Following Corvelo Benz & Rodriguez (2023), we formalize the utility function $u(T, Y)$ as follows:

$$u(1, 1) > u(1, 0), u(1, 1) > u(0, 1), u(0, 0) > u(1, 0), u(0, 0) \geq u(0, 1). \quad (1)$$

**Expertise disparity (ED) and utility disparity (UD).** Assume there are $|\mathcal{S}|$ distinct human decision-makers groups categorized by a sensitive attribute $S \in \mathcal{S}$. To enable statistical quantification, for any given AI advice $a$, we measure the expertise disparity (ED) between the $i$-th and $j$-th human decision-makers groups by calculating the likelihood disparities of the truth labels between different groups, despite human decision-makers having identical confidence $h$ (which reflects the different expertise abilities to estimate the likelihood of the truth label correctly) as follows,

$$ED = P(Y = 1 | f_A(z) = a, z \in \mathcal{Z}_{h,s_i}) - P(Y = 1 | f_A(z) = a, z \in \mathcal{Z}_{h,s_j}), \quad (2)$$

where $\mathcal{Z}_{h,s_i} = \{(x, h, s) | f_H(x) = h, S = s_i\}$ represents the subset of decisions for group $s_i$ with human decision-makers confidence $h$. Human decision-makers are referred to as expertise-heterogeneous if $ED \neq 0$. This issue may not arise in calibration scenarios involving only predictors (Hebert-Johnson et al., 2018); instead, it originates from human behavior, which is unique to AI-assisted decision-making context. For example, specific human subgroups may exhibit overconfidence in likelihood estimation. To mitigate the unfair utility arising from heterogeneous expertise, we aim for the utility to be equal across different human decision-maker groups despite their expertise disparities, that is, for any groups $i, j \in |\mathcal{S}|$ and for any $h, a \in [0, 1]$, the utility disparity (UD) between group $i$ and $j$ is expected to approach 0 as follows,

$$UD = \frac{\sum\limits_{i,j \in \{1,\ldots,|\mathcal{S}|\}, j < i} \left| \mathbb{E}_\pi \left[ u(T, Y) | f_A(z) = a, z \in \mathcal{Z}_{h,s_i} \right] - \mathbb{E}_\pi \left[ u(T, Y) | f_A(z) = a, z \in \mathcal{Z}_{h,s_j} \right] \right|}{\binom{|S|}{2}} \to 0. \quad (3)$$

Specifically, in scenarios involving binary sensitive attributes, the Eq. 3 can be simplified to:

$$UD = |\mathbb{E}_\pi \left[ u(T, Y) | f_A(z) = a, z \in \mathcal{Z}_{h,1} \right] - \mathbb{E}_\pi \left[ u(T, Y) | f_A(z) = a, z \in \mathcal{Z}_{h,0} \right]| \to 0. \quad (4)$$

**Monotone.** When the human decision-makers act rationally, increasing human decision-makers confidence $h$ and AI confidence $a$ raises the probability of human decision-makers making positive final decisions (Corvelo Benz & Rodriguez, 2023).

**Assumption 2.1.** *(Monotone Decision Policy in AI-Assisted Decision Making) Assume that human decision-makers are rational. The decision policy is monotone, meaning that for any AI confidence $a_1$ and $a_2$, and any human decision-makers confidence $h_1$ and $h_2$, if $a_1 \leq a_2$ and $h_1 \leq h_2$, then,*

$$P(T = 1|h_1, a_1) \leq P(T = 1|h_2, a_2), \ where \ P(T = 1) = \pi(h, a). \tag{5}$$

# 3 CAN FAIR UTILITY BE ACHIEVED IN AI-ASSISTED DECISION-MAKING?

## 3.1 FAILURE TO ENSURE FAIR UTILITY UNDER $\alpha$-CALIBRATION

When the AI model produce confidence estimates that accurately represent the distribution of truth labels, it achieves perfect calibration (Pakdaman Naeini et al., 2015; Yin et al., 2019; Zhang et al., 2020). We adopt the statistical notion of $\alpha$-calibration introduced by Hebert-Johnson et al. (2018), which transitions from approximate calibration to perfect calibration by adjusting the hyperparameter $\alpha$ from 1 to 0.

**Definition 3.1.** *($\alpha_y$-Calibration) An AI system with a confidence function $f_A : \mathcal{Z} \rightarrow [0, 1]$ where $\mathcal{Z} = \{\mathcal{X}, \mathcal{H}, \mathcal{S}\}$ satisfies $\alpha_y$-calibration with respect to $\mathcal{Z}$ if there exists $\mathcal{Z}' \subset \mathcal{Z}$ with $|\mathcal{Z}'| \geq (1 - \alpha_y) \cdot |\mathcal{Z}|$, such that for any AI confidence $a \in [0, 1]$, it holds that:*

$$|P(Y = 1 \mid f_A(z) = a, z \in \mathcal{Z}') - a| \leq \alpha_y. \tag{6}$$

The Definition 3.1 bounds the proportion of samples where the difference between the AI confidence and the positive label likelihood exceeds $\alpha_y$ to be less than $\alpha_y$. When the AI confidence is perfectly calibrated ($\alpha_y \rightarrow 0$), it implies that, for the entire sample space $\mathcal{Z}$, the AI confidence $f_A$ aligns exactly with the likelihood of the positive label. Based on this definition, we present the utility disparity under calibration in Theorem 3.2.

**Theorem 3.2.** *(Utility disparity under calibration (Proof in Appendix A.2)) For the AI-assisted decision-making under utility function $u(T, Y)$ in Eq. 1 and the human decision-makers with any monotone AI-assisted decision policy $\pi \in \Pi(H, A)$, such that while AI confidence function $f_A$ is perfectly calibrated, the utility disparity is given by:*

$$|\mathbb{E}_\pi [u(T,Y)|f_A(z) = a, z \in \mathcal{Z}_{h,1}] - \mathbb{E}_\pi [u(T,Y)|f_A(z) = a, z \in \mathcal{Z}_{h,0}]|$$
$$= \mathbf{Q} \cdot |P(Y = 1|f_A(z) = a, z \in \mathcal{Z}_{h,1}) - P(Y = 1|f_A(z) = a, z \in \mathcal{Z}_{h,0})|, \tag{7}$$

*where $\mathbf{Q}$ can be any value in the range $[0, \max (u(0, 0) - u(0, 1), u(1, 1) - u(1, 0))]$.*

Based on the Theorem 3.2, for any monotone decision policies $\pi(h, a)$, attaining optimal fairness necessitates that for any $h, a \in [0, 1]$, $P(Y = 1|f_A(z) = a, z \in \mathcal{Z}_{h,1}) \equiv P(Y = 1|f_A(z) = a, z \in \mathcal{Z}_{h,0})$. However, it is non-trivial to achieve in practice: Consider a disease diagnosis scenario involving two groups of human decision-makers: experts ($S = 1$) and general practitioners ($S = 0$) working under AI assistance. Suppose both groups diagnose patients as having the disease with a human decision-maker confidence level of $h = 0.9$. Due to differences in expertise, there may be a disparity in the true probability that the patients actually have the disease, with the higher-expertise group showing a higher probability: $P(Y = 1|f_A(z) = a, z \in \mathcal{Z}_{0.9,1}) > P(Y = 1|f_A(z) = a, z \in \mathcal{Z}_{0.9,0})$.

## 3.2 FAILURE TO ENSURE FAIR UTILITY UNDER $\alpha_h$-HUMAN-ALIGNMENT

Corvelo Benz & Rodriguez (2023) argued that $\alpha_y$-calibration fails to ensure optimal utility for monotone policies and proposed *human-alignment* as a new calibration objective. They demonstrated that a perfectly human-aligned confidence function guarantees the existence of a monotone policy $\pi$ achieving optimal utility. Inspired by the superior performance of human-alignment, we further analyze the resulting utility disparity.

**Definition 3.3.** *($\alpha_h$-Human-alignment) An AI system with a confidence function $f_A : \mathcal{Z} \rightarrow [0, 1]$ where $\mathcal{Z} = \{\mathcal{X}, \mathcal{H}, \mathcal{S}\}$, satisfies $\alpha_h$-alignment with respect to human decision-maker confidence function $f_H : \mathcal{X} \rightarrow [0, 1]$ if, for any $h \in \mathcal{H}$, there exists $\mathcal{Z}'_h \subset \mathcal{Z}_h$ with $\mathcal{Z}_h =$*

$\{(x, h, s) \mid f_H(x) = h\} \subset \mathcal{Z}$ *and* $|\mathcal{Z}'_h| \geq (1 - \alpha_h/2) \cdot |\mathcal{Z}_h|$, *such that, for any* $0 \leq a_1 \leq a_2 \leq 1$ *and* $0 \leq h_1 \leq h_2 \leq 1$, *it holds that,*

$$P\left(Y = 1 \mid f_A(z) = a_1, z \in \mathcal{Z}'_{h_1}\right) - P\left(Y = 1 \mid f_A(z) = a_2, z \in \mathcal{Z}'_{h_2}\right) \leq \alpha_h, \alpha_h \in [0, 1]. \quad (8)$$

For $h_1$ and $h_2$ satisfying the monotonicity condition $h_1 \leq h_2$, the above definition bounds the violation of monotonicity in the positive label introduced by $f_A$ to at most $\alpha_h/2$ over the sample spaces $\mathcal{Z}_{h_1}$ and $\mathcal{Z}_{h_2}$. However, even if the AI confidence function $f_A$ is perfectly human-aligned ($\alpha_h \to 0$), the monotonic decision policy $\pi$ still be suboptimal in terms of fair utility, as stated in Theorem 3.4.

**Theorem 3.4.** *(Utility disparity under human-alignment (Proof in Appendix A.3)) There exist (infinitely many) AI-assisted decision-making processes with utility function $u(T, Y)$ in Eq. 1 and the human decision-maker with any monotone AI-assisted decision policy $\pi \in \Pi(H, A)$, such that even the AI confidence function $f_A$ is perfectly aligned with the human's, the AI-assisted decision-making still fails to achieve optimal utility fairness. Specifically,*

$$\begin{aligned}
&|\mathbb{E}_\pi\left[u(T, Y)|f_A(z) = a, z \in \mathcal{Z}_{h,1}\right] - \mathbb{E}_\pi\left[u(T, Y)|f_A(z) = a, z \in \mathcal{Z}_{h,0}\right]| \\
&> |\mathbb{E}_{\pi^*}\left[u(T, Y)|f_A(z) = a, z \in \mathcal{Z}_{h,1}\right] - \mathbb{E}_{\pi^*}\left[u(T, Y)|f_A(z) = a, z \in \mathcal{Z}_{h,0}\right]|,
\end{aligned} \quad (9)$$

*where,*

$$\pi^* = \underset{\pi \in \Pi(\mathcal{H}, \mathcal{A})}{\arg\min} |\mathbb{E}_\pi\left[u(T, Y)|f_A(z) = a, z \in \mathcal{Z}_{h,1}\right] - \mathbb{E}_\pi\left[u(T, Y)|f_A(z) = a, z \in \mathcal{Z}_{h,0}\right]|. \quad (10)$$

We analyze that the cause of failure in fairness arises from human-alignment without considering differences in the correctness of human decision-makers' confidence due to heterogeneous expertise. This discrepancy results in differing levels of human-alignment between groups. For groups with weaker alignment, this can lead to utility disadvantages.

## 3.3 INTER-GROUP-ALIGNMENT AND UTILITY DISPARITY UPPER BOUND

Given the limitations of existing calibration methods in ensuring optimal fair utility, we introduce the core concept of *inter-group-alignment* in Definition 3.5. Building on this foundation, we give an upper bound on utility disparity of any AI-assisted decision making process, as in Theorem 3.6.

**Definition 3.5.** *($\alpha_g$-Inter-group-alignment) An AI system with a confidence function $f_A : \mathcal{Z} \to [0, 1]$ where $\mathcal{Z} = [\mathcal{X}, \mathcal{H}, \mathcal{S}]$, satisfies $\alpha_g$-inter-group-alignment if, for any $h \in \mathcal{H}$, there exists $\mathcal{Z}'_h \subset \mathcal{Z}_h$ with $\mathcal{Z}_h = \{(x, h, s) \mid f_H(x) = h\}$ and $|\mathcal{Z}'_h| \geq (1 - \alpha_g/2) \cdot |\mathcal{Z}_h|$. Let $\mathcal{Z}'_{h,s} = \{(x, h, s) \in \mathcal{Z}'_h \mid S = s\}$, the AI confidence is $\alpha_g$-inter-group-alignment if,*

$$\left| P\left(Y = 1 \mid f_A(z) = a, z \in \mathcal{Z}'_{h,1}\right) - P\left(Y = 1 \mid f_A(z) = a, z \in \mathcal{Z}'_{h,0}\right) \right| \leq \alpha_g. \quad (11)$$

Based on Equation 5, given identical AI confidence $a$ and human decision-maker's confidence $h$, human decision-makers will exhibit the same probability of making the final decision, $P(T = 1)$. The definition of $\alpha_g$-inter-group-alignment constrains the distribution of positive label $Y = 1$ to be statistically equal across different human decision-maker groups when $\alpha_g \to 0$. This alignment ensures that human decision-makers within each group achieve statistically similar utilities for making correct decisions.

**Theorem 3.6.** *(Utility disparity upper bound under $\alpha_h$-human-alignment and $\alpha_g$-inter-group-alignment (Proof in Appendix A.4)) For a given AI-assisted decision-making process with a utility function $u(T, Y)$ satisfying Eq. 1 and the human with any monotone AI-assisted decision policy $\pi \in \Pi(H, A)$, if the AI confidence function $f_A$ is $\alpha_h$-human-alignment and satisfies $\alpha_g$-inter-group-alignment, then the utility disparity is bounded by,*

$$\begin{aligned}
&|\mathbb{E}_\pi\left[u(T, Y)|f_A(z) = a, z \in \mathcal{Z}_{h,1}\right] - \mathbb{E}_\pi\left[u(T, Y)|f_A(z) = a, z \in \mathcal{Z}_{h,0}\right]| \\
&\leq (u(1, 1) - u(0, 1) - u(1, 0) + u(0, 0)) \cdot \left(\frac{\alpha_h}{2} + \left(1 - \frac{\alpha_h}{2}\right) \cdot \left(3\alpha_g - \alpha_g^2\right)\right).
\end{aligned} \quad (12)$$

Theorem 3.6 provides a tight upper bound on the utility disparity, which is constrained by both the human-alignment level $\alpha_h$ and inter-group-alignment level $\alpha_g$. This offers valuable insights

into the fairness-aware AI confidence calibration objectives, which seek to align the AI confidence as closely as possible with human decision-makers's, while simultaneously ensuring that human decision-makers from different groups, with identical confidence $h$, receive statistically similar positive label distributions when provided with the same AI confidence $a$. Based on this, we will next outline a practical approach to achieving both calibration objectives simultaneously.

# 4 GROUP-LEVEL CONFIDENCE MULTICALIBRATION FOR SIMULTANEOUS IMPROVEMENT OF UTILITY AND FAIRNESS

Based on Theorem 3.6, we now present the new AI confidence alignment objective to ensure fairer utility across different human decision-maker groups in AI-assisted decision-making, enabling AI-assisted decisions to be more practical for real-world human decision-maker with heterogeneous expertise.

**Corollary 4.1.** *For any AI-assisted decision-making with a utility function $u(T, Y)$ in Eq. 1 and is $\alpha_h$-human-alignment, the upper bound of utility disparity across different human decision-maker groups is minimized when the decision function $f_A$ satisfies perfect inter-group-alignment.*

The above corollary holds as $\left(3\alpha_g - \alpha_g^2\right) \geq 0$ for all $\alpha_g \in [0, 1]$. Consequently, under any human-alignment level $a_h$, the utility disparity upper bound in Theorem 3.6 is minimized when $\alpha_g = 0$. We can further refine the conditions under which the AI-assisted decision-making process provides both optimal utility and fair utility across heterogeneous human decision-maker groups as follows:

**Corollary 4.2.** *For AI-assisted decision-making processes with a utility function $u(T, Y)$ satisfying Eq. 1, if $f_A$ achieves both perfectly human-alignment and perfectly inter-group-alignment, there exist monotone AI-assisted decision policy $\pi \in \Pi(\mathcal{H}, \mathcal{A})$ that simultaneously attains optimal overall utility and fair utility among heterogeneous human decision-maker groups.*

Based on the utility disparity upper bound established in Theorem 3.6, when both $\alpha_h = \alpha_g = 0$, the AI-assisted decision-making system achieves optimal fairness with utility disparity to be 0. In the following, we demonstrate how to simultaneously achieve human-alignment and inter-group-alignment through multicalibration, thereby ensuring that AI-assisted decision-making provides fair utility while guaranteeing optimal utility for all human decision-makers. Multicalibration (Hebert-Johnson et al., 2018) was initially introduced as a measure of algorithm fairness to mitigate discrimination introduced by a predictor's training process.

**Definition 4.3.** *(Multicalibration) Let $\mathcal{C} \subset 2^{\mathbb{Z}}$ be a collection of subsets in domain $\mathbb{Z}$, and let $\alpha \in [0, 1]$. An AI's confidence function $f_A : \mathcal{Z} \to [0, 1]$ is $\alpha$-multicalibrated with respect to $\mathcal{C}$ if, for all $\mathcal{Z} \subset \mathcal{C}$, $f_A$ satisfies $\alpha$-calibration (Definition 3.1) with respect to $\mathcal{Z}$.*

Corvelo Benz & Rodriguez (2023) demonstrated that multicalibration leads to human-alignment. However, as shown in Theorem 3.4 and the experimental results in Figure 2, human-alignment alone does not provide fair utility among heterogeneous human decision-makers. To address this, we introduce *group-level multicalibration* and explain how it ensures fair utility for heterogeneous decision-makers in AI-assisted decision-making, while maintaining overall utility.

**Theorem 4.4.** *($\alpha/2$-Group-level multicalibration leads to $\alpha$-human-alignment and $\alpha$-inter-group-alignment meanwhile (Proof in Appendix A.5)) Let $f_A : \mathcal{Z} \to [0, 1]$ be an AI's confidence function. Suppose in each human decision-maker group $i \in \{1, ..., |S|\}$, $f_A(z)$ is $\alpha/2$-multicalibrated with respect to the collection $\mathcal{C} = \{\mathcal{Z}_{h,s_i}\}_{h \in \mathcal{H}}$ with $\mathcal{Z}_{h,s_i} = \{(x, h, s) | f_H(x) = h, S = s_i\}$, then $f_A$ is both $\alpha$-aligned with respect to the human confidence function $f_H$ and $\alpha$-inter-group aligned across the different human decision-maker groups.*

$\alpha/2$**-Group-level-multicalibration by $\lambda$-discretization.** We present the key steps to achieve $\alpha/2$-group-level multicalibration as follows, with a detailed algorithm provided in Algorithm A.6: For each $\mathcal{Z}_{h,s_i} \in \mathcal{C}$, we apply $\lambda$-discretization (Hebert-Johnson et al., 2018) to the AI's confidence function $f_A$. Specifically, $\lambda$-discretization partitions the $f_A$ confidence interval $[0, 1]$ into $\lfloor 1/\lambda \rfloor$ discrete bins, each with a width of $\lambda$. The centers of these bins are located at $\Lambda = \left\{ \frac{\lambda}{2}, \frac{3\lambda}{2}, \ldots, 1 - \frac{\lambda}{2} \right\}$. The $\lambda$-discretization partitions $\mathcal{Z}_{h,s_i}$ into $\mathcal{Z}_{h,s_i}^j = \{z \in \mathcal{Z}_{h,s_i} | f_A(z) \in [\Lambda[j] - \lambda/2, \Lambda[j] + \lambda/2)\}, j = \{1, ..., \lfloor 1/\lambda \rfloor\}$. We then iteratively update the AI's confidence estimates for all instances $z \in$

$\mathcal{Z}^j_{h,s_i}, j = 1, ..., \lfloor 1/\lambda \rfloor$ as follows,

$$f_A(z) \to f_A(z) + P(Y = 1 | z \in \mathcal{Z}^j_{h,s_i}) - \mathbb{E}\left[f_A(z) | z \in \mathcal{Z}^j_{h,s_i}\right]. \tag{13}$$

This process continues until the following discretized notion of $\widetilde{\alpha}$-multicalibration is satisfied on all discrete partitions as follows,

$$\left| E\left[f_A(z) | z \in \mathcal{Z}^j_{h,s_i}\right] - P(Y = 1 | z \in \mathcal{Z}^j_{h,s_i}) \right| \le \widetilde{\alpha}. \tag{14}$$

After completing the aforementioned discretized $\widetilde{\alpha}$-multicalibration, for $i$-th human decision-maker group, the algorithm proceeds to return a discretized confidence function in $j$-th bin as $f_A(z) = E\left[f_A(z) | z \in \mathcal{Z}^j_{s_i}\right]$. According to (Hebert-Johnson et al., 2018), the discretized confidence function provides $(\widetilde{\alpha}+\lambda)$-multicalibration for each human decision-maker group. Based on Theorem 4.4, to obtain a discretized confidence function $f_A$ that satisfies at least $\alpha$-human-alignment and $\alpha$-inter-group-alignment, it is necessary to ensure that $\widetilde{\alpha} + \lambda \le \alpha/2$ for each group. We further analyze how $\widetilde{\alpha}$ and $\lambda$ impact the efficiency of group-level multicalibration in Appendix A.8.1.

# 5 EXPERIMENTS

## 5.1 SETTINGS

**Dataset.** We utilize a publicly available dataset for human-AI interactions across 4 tasks (Vodrahalli et al., 2022a). In each task, human decision-makers first provide their confidence (used to construct $f_H$). After receiving AI advice (used to construct $f_A$), participants update their final decision confidence (used to construct $\pi(h, a)$). Additionally, the dataset includes basic demographic information of the participants, such as gender, as provided by the crowdsourcing platform. The 4 tasks span different data modalities (visual, text, and tabular) and are sufficiently challenging to ensure that participants can benefit from AI assistance. In the Art (Image) task, participants determine the art period of a painting from two options. In the Cities (Image) task, participants are asked to determine the originating city of an image from a binary choice. In the Sarcasm (Text) task, participants determine if a Reddit text snippet contains sarcasm. In the Census (Tabular) task, participants assess whether an individual earns at least \$50,000 annually based on their demographic information. The human decision-makers are divided into two groups ("Female" as Group $S = 0$ and "Male" as Group $S = 1$). The data are preprocessed to filter out samples with missing information and confounding factors (Appendix A.7.1), resulting in $14,999$ AI-assisted decision-making records from 469 participants overall.

**Hyperparameters.** We configure the hyperparameters as follows: $\widetilde{\alpha} = 0.0001$ and $\lambda = 0.125$, ensuring that the level of group-level multicalibration is approximately $0.125$.

**Decision policy function.** Since the dataset only provides uncalibrated confidences, we evaluate AI-assisted decision performance after calibration by learning the decision policy $\pi(h, a)$ using a multi-layer perceptron (MLP) classifier with one hidden layer of 20 nodes and ReLU activation.

**Experimental setup.** We establish three AI confidence calibration cases for each task: no calibration (before calibration), after multicalibration (Corvelo Benz & Rodriguez, 2023), and after group-level multicalibration. Under each condition, we conduct the following experiments: ❶ Alignment quantification: We evaluate the effectiveness of the proposed group-level multicalibration in achieving human-alignment and inner-group-alignment. ❷ Expected utility and utility disparity of final decision $\pi(h, a)$: We compare the overall expected utility $\mathbb{E}_{\pi(h,a)}\left[u(T, Y)\right]$ and the utility disparity $\mathbb{E}_{\pi(h,a)}\left[u(T, Y) | f_A(z) = a, z \in \mathcal{Z}_{h,1}\right] - \mathbb{E}_{\pi(h,a)}\left[u(T, Y) | f_A(z) = a, z \in \mathcal{Z}_{h,0}\right]$ to evaluate how well group-level multicalibration can improve fair utility across diverse human decision-maker groups and optimal utility simultaneously. ❸ Expected utility and utility disparity of human-only decision $\pi(h)$ and AI-only decision $\pi(a)$: We compare the utility $\mathbb{E}_{\pi(h)}\left[u(T, Y)\right]$ ($\mathbb{E}_{\pi(a)}\left[u(T, Y)\right]$) and utility disparities $\mathbb{E}_{\pi(h)}\left[u(T, Y) | f_A(z) = a, z \in \mathcal{Z}_{h,1}\right] - \mathbb{E}_{\pi(h)}\left[u(T, Y) | f_A(z) = a, z \in \mathcal{Z}_{h,0}\right]$ ($\mathbb{E}_{\pi(a)}\left[u(T, Y) | f_A(z) = a, z \in \mathcal{Z}_{h,1}\right] - \mathbb{E}_{\pi(a)}\left[u(T, Y) | f_A(z) = a, z \in \mathcal{Z}_{h,0}\right]$) to understand why achieving inter-group-alignment by group-level multicalibration supports fair utility across diverse decision maker groups.

Table 1: Alignment evaluation under no calibration, multicalibration, and group-level multicalibration (**Bold** represents the best result, underlined represents the second-best result).

| Experiment | Calibration Methods | | | | | | | | | | | |
|---|---|---|---|---|---|---|---|---|---|---|---|---|
| | None | | | | Multicalibration | | | | Group-level Multicalibration | | | |
| | EAE | MAE | EIAE | MIAE | EAE | MAE | EIAE | MIAE | EAE | MAE | EIAE | MIAE |
| 1 | **0.0006** | 0.0576 | 0.0658 | 0.2701 | **0.0006** | **0.0323** | 0.0709 | 0.3760 | 0.0016 | 0.0875 | **0.0110** | **0.0970** |
| 2 | 0.0045 | 0.2239 | 0.0626 | 0.2599 | **0.0000** | **0.0000** | 0.0289 | 0.3912 | 0.0005 | 0.0465 | **0.0049** | **0.0790** |
| 3 | **0.0001** | **0.0134** | 0.0449 | 0.2049 | 0.0006 | 0.0590 | 0.0250 | 0.1881 | 0.0007 | 0.0606 | **0.0064** | **0.0674** |
| 4 | 0.0088 | 0.2985 | 0.0607 | 0.3651 | **0.0002** | **0.0195** | 0.0316 | 0.2807 | 0.0003 | 0.0421 | **0.0054** | **0.1386** |

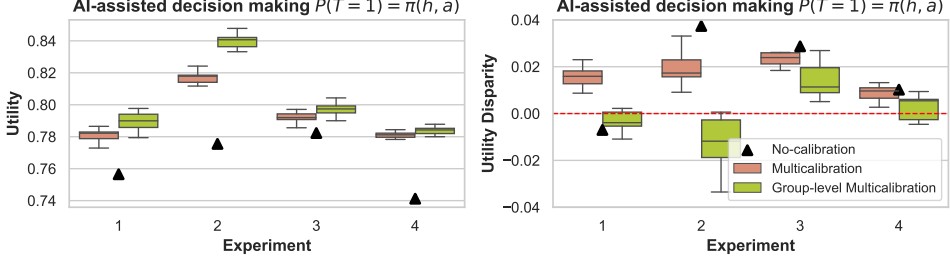

Figure 2: Statistics of utility and utility disparity over 100 experiments, where the final decision $P(T = 1) = \pi(h, a)$ is made by human with AI assistance. The AI confidence is either uncalibrated or calibrated using multicalibration and group-level multicalibration, respectively.

**Evaluation metric.** We use accuracy to evaluate the decision utility and, naturally, evaluate the fair utility to diverse human decision-maker groups by measuring accuracy disparities as follows,

$$\text{Disp} = \mathbb{E}\left[\mathbf{1}(T = Y)|S = 1\right] - \mathbb{E}\left[\mathbf{1}(T = Y)|S = 0\right]. \tag{15}$$

Human-alignment is measured through two primary metrics: the expected alignment error (EAE) and the maximum alignment error (MAE) (Corvelo Benz & Rodriguez, 2023).

$$\text{EAE} = \max\left(0, \frac{1}{N} \cdot \sum_{i \leq i', j \leq j'} \left[ P\left(Y = 1 \mid z \in \mathcal{Z}_i^j\right) - P\left(Y = 1 \mid z \in \mathcal{Z}_{i'}^{j'}\right) \right] \right). \tag{16}$$

$$\text{MAE} = \max\left(0, \max_{i \leq i', j \leq j'} \left( P\left(Y = 1 \mid z \in \mathcal{Z}_i^j\right) - P\left(Y = 1 \mid z \in \mathcal{Z}_{i'}^{j'}\right) \right) \right). \tag{17}$$

Following the discretization process (Corvelo Benz & Rodriguez, 2023), we develop metrics for assessing inter-group-alignment: the expected inter-group-alignment error (EIAE) and the maximum inter-group-alignment error (MIAE).

$$\text{EIAE} = \frac{1}{N} \cdot \sum_{i,j} \left| P\left(Y = 1 \mid z \in \mathcal{Z}_{i,1}^j\right) - P\left(Y = 1 \mid z \in \mathcal{Z}_{i,0}^j\right) \right|. \tag{18}$$

$$\text{MIAE} = \max_{i,j} \left( \left| P\left(Y = 1 \mid z \in \mathcal{Z}_{i,1}^j\right) - P\left(Y = 1 \mid z \in \mathcal{Z}_{i,0}^j\right) \right| \right), \tag{19}$$

where $Z_{i,0}^j$ and $Z_{i,1}^j$ contain samples from the groups $S = 0$ and $S = 1$, respectively, located in the $(i, j)$-th cell of the grid formed by the discretization of human confidence and AI confidence. The discretization details is in the Appendix A.7.2. The limitation of metrics based on discretization lies in the finite number of discrete intervals, which may fail to capture alignment across the entire continuous confidence space accurately. However, significant variations in the metrics can reflect variations in alignment more accurately. In such cases, the influence of unmeasured alignment on the overall results is reduced.

## 5.2 RESULTS

The key takeaway from the experimental results presented in Table 5.1 is an evaluation of the proposed group-level multicalibration method's effectiveness in ensuring both human-alignment and inter-group-alignment. In the Cities and Centus tasks, both multicalibration and group-level multicalibration significantly reduce EAE and MAE compared to the uncalibrated case, indicating more substantial alignment with human decision-maker's confidence. While the uncalibrated model yields

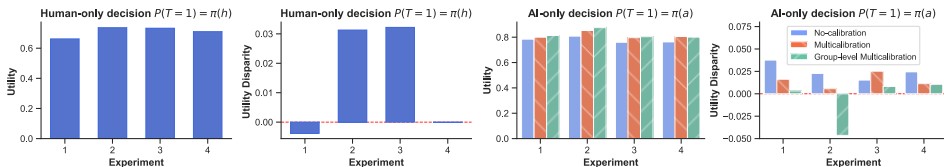

Figure 3: The utility and utility disparity where the final decision is made by human-only $P(T = 1) = \pi(h)$ or AI-only $P(T = 1) = \pi(a)$.

the best performance for the Art and Sarcasm tasks, the differences in EAE and MAE across all calibration methods are subtle, with the maximum variation being 0.001. As previously noted, discrete statistical metrics may lose precision when capturing such minor differences due to the limitations of discrete interval choices. Therefore, multicalibration and group-level multicalibration can be regarded as demonstrating comparable levels of human-alignment to the uncalibrated model in these tasks. This observation becomes particularly evident in subsequent experiments, where the actual utility, measured by accuracy, demonstrates a significant improvement under both multicalibration and group-level multicalibration across all tasks compared to the uncalibrated model. When evaluating inter-group-alignment using the EIAE and MIAE metrics, group-level multicalibration consistently achieves the best performance across all tasks, significantly reducing EIAE and MIAE compared to uncalibrated cases and multicalibration.

In Figure 2, the key takeaways are: 1) multicalibration fails to improve fairness and, in some cases, exacerbates fairness issues, as shown in task 1, where the utility disparity after multicalibration is worse than in the uncalibrated scenario; 2) group-level multicalibration consistently outperforms both the uncalibrated case and multicalibration across all tasks, improving decision utility and reducing utility disparity across different human decision-maker groups. Using an MLP-based decision model, we report the distribution of final decisions over 100 trials with random seeds $(0 - 99)$. The uncalibrated case shows no variance, as its decisions are fixed by the dataset.

In Figure 3, we provide the utility and utility disparity when final decisions are made solely by human decision-makers and AI independently. The key observation is that group-level multicalibration adjusts the AI's confidence to mitigate utility disparity caused by human-only decisions, either by reducing the disparity or by creating an offsetting disparity with the opposite sign. This capability is absent in multicalibration, which in some cases (e.g., the Sarcasm task) can even worsen utility disparity compared to human-only decisions. This experiment highlights the advantage of group-level multicalibration in promoting fairer utility by effectively adjusting AI confidence.

## 6 DISCUSSION

### 6.1 RELATED WORK

AI-assisted decision-making involves human decision-makers taking advice from AI systems. To establish a productive working relationship between human decision-makers and AI, the AI model is expected to provide an interpretable and explainable decision-making process. A direct approach involves AI systems providing confidence for their predictions (Bhatt et al., 2021; Steyvers & Kumar, 2024; Ma et al., 2023; Zhang et al., 2020), i.e., the likelihood of classification outcomes. AI confidence helps decision-makers calibrate their trust in the AI and appropriately apply AI knowledge to make final decisions, especially in cases where the AI model is likely to perform poorly. To enhance human decision-maker's comprehension of AI prediction uncertainty, AI model confidence is primarily calibrated to reflect the probabilities of classification correctness (Hebert-Johnson et al., 2018; Guo et al., 2017; Zhao et al., 2021). However, experimental evidence by Vodrahalli et al. (2022b) indicated that AI models, when perceived as more confident than they actually are—rather than being well-calibrated—can enhance the accuracy of final decisions made by human decision-makers after considering AI advice. The work most closely related to ours is Corvelo Benz & Rodriguez (2023), which conducted a systematic theoretical analysis of scenarios where well-calibrated AI confidence may lead to suboptimal utility for rational decision-makers. They also introduced the concept of AI confidence human-alignment, enabling rational decision-makers to achieve optimal utility. Previous works in AI-assisted decision making assume that human decision-makers are ho-

mogeneous, overlooking heterogeneity (Rambachan, 2024; Rambachan et al., 2024; De-Arteaga et al., 2024) in their expertise. Since heterogeneity in expertise may stem from historical inequities in education and access to resources, mitigating the resulting utility disparity is critical for promoting societal welfare. When AI assistance is used as a new information resource, failing to account for expertise heterogeneity can exacerbate utility disparities among decision-makers, exacerbating societal inequities. Distinguished from Corvelo Benz & Rodriguez (2023), our work addresses a novel and previously unexplored dimension of AI-assisted decision-making: ensuring equitable utility for human decision-makers with varying levels of expertise. Furthermore, we contribute to a solid theoretical framework for analyzing and mitigating fairness issues in AI-assisted decision-making. Different from the commonly used algorithmic fairness in the fair machine learning area, which mainly aims to ensure unbiased outcomes for individuals being decided upon (e.g., patients), our focus is on ensuring fairness for the decision-makers (e.g., doctors). Pleiss et al. (2017) investigated the compatibility between calibration and Equalized Odds. Beyond differences in application contexts—where their focus is on predictor scenarios rather than AI-assistance—our work achieve human-alignment and inter-group alignment in a compatible manner, also targeting a distinct fairness concept. This fairness concept shares some similarities with accuracy disparity in centralized model training (Chi et al., 2021) and egalitarian fairness in decentralized learning (Donahue & Kleinberg, 2023), but it requires fundamentally different solving methods tailored to the context of AI-assisted decision-making.

### 6.2 SCOPE AND FUTURE WORK

**Multi-class and multi-groups.** Our theoretical results regarding the existence and mitigation for the unfair utility are currently limited to binary classification tasks and binary human decision-maker groups. The proofs presented in our work are modular, and it is possible that illuminating properties exist in the broader context of AI-assisted decision-making processes. However, extending the theoretical analysis directly to multi-class classification and multiple human decision-maker groups presents several challenges. The first challenge is identifying more natural properties that utility functions may satisfy in multi-class classification. For example, the utility of diagnosing a patient with Type 1 diabetes ($Y = 1$) as Type 2 diabetes ($T = 2$) may yield utility than diagnosing them as disease-free ($T = 0$); that is, $u(T = 2, Y = 1) > u(T = 0, Y = 1)$, which may be more complex than binary classification. Second, with multiple human groups, there may be alternative forms of fair utility, such as focusing on the max-min gap or the standard deviation of the utilities across all human decision-maker groups, reflecting different social welfare objectives. This raises questions about human decision-makers behavior analysis and their preferences regarding various notions of fair utility, deserving further exploration in subsequent research.

**Fairness metrics.** In this work, we use utility disparity for measuring fair utility, aligning with the decision-makers' primary goal of making more accurate decisions with AI assistance (Steyvers & Kumar, 2024). The concept of utility disparity is also evident in centralized learning (Chi et al., 2021) and decentralized learning (Donahue & Kleinberg, 2023). While other fairness metrics, such as demographic parity and equalized odds (Mehrabi et al., 2021), emphasize the equality of positive or true positive outcomes, these metrics primarily relate to individuals being judged (e.g., patients) and diverge from the fair utility objective for decision-makers (e.g., doctors). Nonetheless, exploring diverse fairness concepts remains an interesting avenue for future work.

## 7 CONCLUSIONS

In this work, we have systematically analyzed the issue of unfair utility that arises when human decision-makers with heterogeneous expertise engage in AI-assisted decision-making. We have identified that rational decision-makers incorporating AI confidence may not achieve equal utility under existing AI confidence calibration criteria. To address this issue, we have introduced a novel confidence calibration criterion, inter-group-alignment, which, when combined with human-alignment, establishes an upper bound on utility disparities. Building on this foundation, we have proposed group-level multicalibration to enable AI confidence to achieve both human-alignment and inter-group-alignment simultaneously. Experiments conducted on real datasets have thoroughly evaluated the effectiveness of our new AI confidence calibration criterion and approach in providing optimal and fair utility across heterogeneous human decision-maker groups.

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

# A APPENDIX

## A.1 PRE-LEMMAS

**Lemma A.1.** *If the utility function $u$ satisfies Eq. 1 and the distribution of $Y = 1$ satisfies $P(Y = 1|S = 1) > P(Y = 1|S = 0)$, then a trivial policy $\pi$ that always decides $T = 1$ will consistently result in a positive utility disparity, while a trivial policy that always decides $T = 0$ will consistently result in a negative utility disparity.*

$$\mathbb{E}_{Y \in P}\left[u(1, Y)|S = 1\right] - \mathbb{E}_{Y \in P}\left[u(1, Y)|S = 0\right] > 0, \tag{20}$$

$$\mathbb{E}_{Y \in P}\left[u(0, Y)|S = 1\right] - \mathbb{E}_{Y \in P}\left[u(0, Y)|S = 0\right] \le 0. \tag{21}$$

**Lemma A.2.** *If the utility function $u$ satisfies Eq. 1 and the distribution of $Y = 1$ satisfies $P(Y = 1|S = 1) < P(Y = 1|S = 0)$, then a trivial policy $\pi$ that always decides $T = 1$ will consistently result in a negative utility disparity, while a trivial policy that always decides $T = 0$ will consistently result in a positive utility disparity.*

$$\mathbb{E}_{Y \in P}\left[u(1, Y)|S = 1\right] - \mathbb{E}_{Y \in P}\left[u(1, Y)|S = 0\right] < 0, \tag{22}$$

$$\mathbb{E}_{Y \in P}\left[u(0, Y)|S = 1\right] - \mathbb{E}_{Y \in P}\left[u(0, Y)|S = 0\right] \ge 0. \tag{23}$$

**Proof.** *For Lemma A.1: As $u(1, 1) > u(1, 0)$ and $u(0, 0) \ge u(0, 1)$, when $P(Y = 1|S = 1) > P(Y = 1|S = 0)$, we have*

$$\begin{aligned}
&\mathbb{E}_{Y \in P}\left[u(1, Y)|S = 1\right] - \mathbb{E}_{Y \in P}\left[u(1, Y)|S = 0\right] \\
&= P(Y = 1|S = 1) \cdot u(1, 1) + (1 - P(Y = 1|S = 1)) \cdot u(1, 0) \\
&\quad - P(Y = 1|S = 0) \cdot u(1, 1) - (1 - P(Y = 1|S = 0)) \cdot u(1, 0) \\
&= (P(Y = 1|S = 1) - P(Y = 1|S = 0)) \cdot (u(1, 1) - u(1, 0)) > 0.
\end{aligned} \tag{24}$$

$$\begin{aligned}
&\mathbb{E}_{Y \in P}\left[u(0, Y)|S = 1\right] - \mathbb{E}_{Y \in P}\left[u(0, Y)|S = 0\right] \\
&= P(Y = 1|S = 1) \cdot u(0, 1) + (1 - P(Y = 1|S = 1)) \cdot u(0, 0) \\
&\quad - P(Y = 1|S = 0) \cdot u(0, 1) - (1 - P(Y = 1|S = 0)) \cdot u(0, 0) \\
&= (P(Y = 1|S = 1) - P(Y = 1|S = 0)) \cdot (u(0, 1) - u(0, 0)) \le 0.
\end{aligned} \tag{25}$$

*For Lemma A.2: Similarly, when $P(Y = 1|S = 1) < P(Y = 1|S = 0)$, we have,*

$$\begin{aligned}
&\mathbb{E}_{Y \in P}\left[u(1, Y)|S = 1\right] - \mathbb{E}_{Y \in P}\left[u(1, Y)|S = 0\right] \\
&= (P(Y = 1|S = 1) - P(Y = 1|S = 0)) \cdot (u(1, 1) - u(1, 0)) < 0.
\end{aligned} \tag{26}$$

$$\mathbb{E}_{Y \in P}\left[u(0, Y)|S = 1\right] - \mathbb{E}_{Y \in P}\left[u(0, Y)|S = 0\right]$$
$$= \left(P(Y = 1|S = 1) - P(Y = 1|S = 0)\right) \cdot \left(u(0, 1) - u(0, 0)\right) \geq 0. \tag{27}$$

$\square$

### A.2 PROOF OF THEOREM 3.2.

**Theorem 3.2** (Utility disparity under calibration) For the AI-assisted decision-making under utility function $u(T, Y)$ in Eq. 1 and the human decision-makers with any monotone AI-assisted decision policy $\pi \in \Pi(H, A)$, such that while AI confidence function $f_A$ is perfectly calibrated, the utility disparity is given by:

$$\left|\mathbb{E}_{\pi}\left[u(T, Y)|f_A(z) = a, z \in \mathcal{Z}_{h,1}\right] - \mathbb{E}_{\pi}\left[u(T, Y)|f_A(z) = a, z \in \mathcal{Z}_{h,0}\right]\right|$$
$$= \mathbf{Q} \cdot \left|P(Y = 1|f_A(z) = a, z \in \mathcal{Z}_{h,1}) - P(Y = 1|f_A(z) = a, z \in \mathcal{Z}_{h,0})\right|. \tag{28}$$

where $\mathbf{Q}$ can be any value in the range $[0, \max\left(u(0, 0) - u(0, 1), u(1, 1) - u(1, 0)\right)]$.

***Proof.*** *According to the law of total expectation, and the final decision $P(T = 1) = \pi(h, a)$ independent of the sensitive attribute $S$ (consistent with reality that different human decision-makers make decisions based solely on their own confidence and those of the AI (Corvelo Benz & Rodriguez, 2023)), the expected utility disparity can be formulated as follows:*

$$\left|\mathbb{E}_{\pi}\left[u(T, Y)|f_A(z) = a, z \in \mathcal{Z}_{h,1}\right] - \mathbb{E}_{\pi}\left[u(T, Y)|f_A(z) = a, z \in \mathcal{Z}_{h,0}\right]\right|$$
$$= \left|P(Y = 1|f_A(z) = a, z \in \mathcal{Z}_{h,1}) - P(Y = 1|f_A(z) = a, z \in \mathcal{Z}_{h,0})\right|$$
$$\cdot \left|(u(1, 1) - u(1, 0) - u(0, 1) + u(0, 0)) \cdot P_{\pi}(T = 1|f_A(z) = a, z \in \mathcal{Z}_h) - (u(0, 0) - u(0, 1))\right|. \tag{29}$$

*To prove the above equation, let's first look at $\mathbb{E}_{\pi}\left[u(T, Y)|f_A(z) = a, z \in \mathcal{Z}_{h,1}\right]$.*

$$\mathbb{E}_{\pi}\left[u(T, Y)|f_A(z) = a, z \in \mathcal{Z}_{h,1}\right]$$
$$= \mathbb{E}\left[u(1, Y)|f_A(z) = a, z \in \mathcal{Z}_{h,1}\right] \cdot P_{\pi}(T = 1|f_A(z) = a, z \in \mathcal{Z}_h)$$
$$+ \mathbb{E}\left[u(0, Y)|f_A(z) = a, z \in \mathcal{Z}_{h,1}\right] \cdot \left[1 - P_{\pi}(T = 1|f_A(z) = a, z \in \mathcal{Z}_h)\right]$$
$$= P_{\pi}(T = 1|f_A(z) = a, z \in \mathcal{Z}_h)$$
$$\cdot \left[\mathbb{E}\left[u(1, Y)|f_A(z) = a, z \in \mathcal{Z}_{h,1}\right] - \mathbb{E}\left[u(0, Y)|f_A(z) = a, z \in \mathcal{Z}_{h,1}\right]\right]$$
$$+ \mathbb{E}\left[u(0, Y)|f_A(z) = a, z \in \mathcal{Z}_{h,1}\right]$$

*Similarly, $\mathbb{E}_{\pi}\left[u(T, Y)|f_A(z) = a, z \in \mathcal{Z}_{h,0}\right]$ can be formulated as followed.*

$$\mathbb{E}_{\pi}\left[u(T, Y)|f_A(z) = a, z \in \mathcal{Z}_{h,0}\right]$$
$$= P_{\pi}(T = 1|f_A(z) = a, z \in \mathcal{Z}_h)$$
$$\cdot \left[\mathbb{E}\left[u(1, Y)|f_A(z) = a, z \in \mathcal{Z}_{h,0}\right] - \mathbb{E}\left[u(0, Y)|f_A(z) = a, z \in \mathcal{Z}_{h,0}\right]\right]$$
$$+ \mathbb{E}\left[u(0, Y)|f_A(z) = a, z \in \mathcal{Z}_{h,0}\right]$$

*Therefore, this equation $\left|\mathbb{E}_{\pi}\left[u(T, Y)|f_A(z) = a, z \in \mathcal{Z}_{h,1}\right] - \mathbb{E}_{\pi}\left[u(T, Y)|f_A(z) = a, z \in \mathcal{Z}_{h,0}\right]\right|$ can be expanded into the following form.*

$$\left|\mathbb{E}_{\pi}\left[u(T, Y)|f_A(z) = a, z \in \mathcal{Z}_{h,1}\right] - \mathbb{E}_{\pi}\left[u(T, Y)|f_A(z) = a, z \in \mathcal{Z}_{h,0}\right]\right|$$
$$= \left|P(Y = 1|f_A(z) = a, z \in \mathcal{Z}_{h,1}) - P(Y = 1|f_A(z) = a, z \in \mathcal{Z}_{h,0})\right|$$
$$\cdot \left|[u(1, 1) - u(1, 0) - u(0, 1) + u(0, 0)] \cdot P_{\pi}(T = 1|f_A(z) = a, z \in \mathcal{Z}_h) - [u(0, 0) - u(0, 1)]\right|.$$

*Let*

$$\mathbf{Q} = \left|[u(1, 1) - u(1, 0) - u(0, 1) + u(0, 0)] \cdot P_{\pi}(T = 1|f_A(z) = a, z \in \mathcal{Z}_h) - [u(0, 0) - u(0, 1)]\right|$$
$$\sim [0, \max\left(u(0, 0) - u(0, 1), u(1, 1) - u(1, 0)\right)].$$

*Given that the decision policy $\pi(h, a)$ is monotone, it holds that,*

$$\exists h, a : P_{\pi}(T = 1|f_A(z) = a, z \in \mathcal{Z}_h) \neq \frac{u(0, 0) - u(0, 1)}{u(1, 1) - u(1, 0) - u(0, 1) + u(0, 0)} \rightarrow \mathbf{Q} \neq 0. \tag{30}$$

*Therefore, to ensure the fair utility of diverse human decision-maker groups, there should have,*

$$P(Y = 1|f_A(z) = a, z \in \mathcal{Z}_{h,1}) - P(Y = 1|f_A(z) = a, z \in \mathcal{Z}_{h,0}) \equiv 0. \tag{31}$$

*However, perfect calibration alone does not guarantee the above constraint. According to Definition 3.1, when $f_A$ is perfectly calibrated, it holds that,*

$$\begin{aligned} &P(\mathcal{Y} = 1 \mid f_A(z) = a, z \in \mathcal{Z}_h) \\ &= P(Y = 1|f_A(z) = a, z \in \mathcal{Z}_{h,1}) \cdot P(S = 1|f_A(z) = a, z \in \mathcal{Z}_h) \\ &+ P(Y = 1|f_A(z) = a, z \in \mathcal{Z}_{h,0}) \cdot P(S = 0|f_A(z) = a, z \in \mathcal{Z}_h) = a \end{aligned} \tag{32}$$

*We use an example to illustrate the case where the existence of perfect calibration satisfying Eq. 32 does not guarantee Eq. 31: consider a scenario where the sensitive group proportions satisfy $P(S = 1|f_A(z) = a, z \in \mathcal{Z}_h) = 0.75$ and $P(S = 0|f_A(z) = a, z \in \mathcal{Z}_h) = 0.25$, with $P(Y = 1|f_A(z) = a, z \in \mathcal{Z}_{h,1}) = 2a/3$ and $P(Y = 1|f_A(z) = a, z \in \mathcal{Z}_{h,0}) = 2a$. This setup satisfies perfect calibration as:*

$$P(Y = 1 \mid f_A(z) = a, z \in \mathcal{Z}_h) = \left(\frac{2a}{3}\right) \cdot 0.75 + (2a) \cdot 0.25 = a. \tag{33}$$

*However,*

$$P(Y = 1|f_A(z) = a, z \in \mathcal{Z}_{h,1}) - P(Y = 1|f_A(z) = a, z \in \mathcal{Z}_{h,0}) \neq 0. \tag{34}$$

$\square$

### A.3 PROOF OF THEOREM 3.4

**Theorem 3.4** (Utility disparity under human-alignment) There exist (infinitely many) AI-assisted decision-making processes with utility function $u(T, Y)$ in Eq. 1 and the human decision-maker with any monotone AI-assisted decision policy $\pi \in \Pi(H, A)$, such that while the AI confidence function $f_A$ is perfect human-alignment, the AI-assisted decision making is suboptimal with respect to fair utility. Specifically,

$$\begin{aligned} &|\mathbb{E}_\pi [u(T,Y)|f_A(z) = a, z \in \mathcal{Z}_{h,1}] - \mathbb{E}_\pi [u(T,Y)|f_A(z) = a, z \in \mathcal{Z}_{h,0}]| \\ &> |\mathbb{E}_{\pi^*} [u(T,Y)|f_A(z) = a, z \in \mathcal{Z}_{h,1}] - \mathbb{E}_{\pi^*} [u(T,Y)|f_A(z) = a, z \in \mathcal{Z}_{h,0}]| . \end{aligned} \tag{35}$$

where,

$$\pi^* = \underset{\pi \in \Pi(\mathcal{H},\mathcal{A})}{arg\min} |\mathbb{E}_\pi [u(T,Y)|f_A(z) = a, z \in \mathcal{Z}_{h,1}] - \mathbb{E}_\pi [u(T,Y)|f_A(z) = a, z \in \mathcal{Z}_{h,0}]| . \tag{36}$$

***Proof.*** *We first define $\bar{a}$, which represents the smallest AI system's confidence value for given confidence level $h$, such that,*

$$\bar{a} = \min \left\{ a \in \mathcal{A} \mid P(Y = 1 \mid f_A(z) = a, z \in \mathcal{Z}_{h,1}) - P(Y = 1 \mid f_A(z) = a, z \in \mathcal{Z}_{h,0}) > 0 \right\}. \tag{37}$$

*We demonstrate through the following four cases that there are infinitely many AI-assisted decision-making processes where, despite the AI confidence being human-aligned, the AI-assisted system fails to achieve optimal utility disparity.*

***Case 1.*** *For any confidence $[h_1, a_1]$ with $a_1 < \bar{a}_1$, according to Eq. 37, it holds that,*

$$P(Y = 1 \mid f_A(z) = a_1, z \in \mathcal{Z}_{h_1,1}) - P(Y = 1 \mid f_A(z) = a_1, z \in \mathcal{Z}_{h_1,0}) \leq 0. \tag{38}$$

*Furthermore, there exists another $[h_2, a_2]$, where $a_2 > \max(\bar{a}_2, a_1)$ and $h_2 > h_1$ such that,*

$$P(Y = 1 \mid f_A(z) = a_2, z \in \mathcal{Z}_{h_2,1}) - P(Y = 1 \mid f_A(z) = a_2, z \in \mathcal{Z}_{h_2,0}) > 0. \tag{39}$$

*In the case where $f_A$ is $\alpha_h$-alignment with respect to $f_H$, according to Definition 3.3, for any $h \in \mathcal{H}$, there exists $\mathcal{Z}'_h \subset \mathcal{Z}_h$ with $\mathcal{Z}_h = \{(x, h, s) | f_H(x) = h\} \subset \mathcal{Z}$ and $|\mathcal{Z}'_h| \geq (1 - \alpha_h/2) \cdot |\mathcal{Z}_h|$ such that,*

$$P(Y = 1|a_1, z \in \mathcal{Z}'_{h_1}) - P(Y = 1|a_2, z \in \mathcal{Z}'_{h_2}) = \alpha^* < \alpha, \alpha = \max(0, \alpha^*). \tag{40}$$

Based on the law of total probability, the Eq. 40 can be expanded as follows:

$$
\begin{aligned}
& P(Y = 1 | a_1, z \in \mathcal{Z}'_{h_1,1}) \cdot P(S = 1 | a_1, z \in \mathcal{Z}'_{h_1}) \\
& + P(Y = 1 | a_1, z \in \mathcal{Z}'_{h_1,0}) \cdot P(S = 0 | a_1, z \in \mathcal{Z}'_{h_1}) \\
& = P(Y = 1 | a_2, z \in \mathcal{Z}'_{h_2,1}) \cdot P(S = 1 | a_2, z \in \mathcal{Z}'_{h_2}) \\
& + P(Y = 1 | a_2, z \in \mathcal{Z}'_{h_2,0}) \cdot P(S = 0 | a_2, z \in \mathcal{Z}'_{h_2}) + \alpha^*.
\end{aligned}
\tag{41}
$$

We can quantify the utility disparity gap under different confidence settings when the decision-maker consistently chooses $T = 1$ as follows:

$$
\begin{aligned}
& \Big( \mathbb{E}\left[u(1, Y) | a_1, z \in \mathcal{Z}'_{h_1,1}\right] - \mathbb{E}\left[u(1, Y) | a_1, z \in \mathcal{Z}'_{h_1,0}\right] \Big) \\
& - \Big( \mathbb{E}\left[u(1, Y) | a_2, z \in \mathcal{Z}'_{h_2,1}\right] - \mathbb{E}\left[u(1, Y) | a_2, z \in \mathcal{Z}'_{h_2,0}\right] \Big) \\
& = \Big( u(1, 1) - u(1, 0) \Big) \cdot \Delta_1.
\end{aligned}
\tag{42}
$$

where,

$$
\begin{aligned}
\Delta_1 = & P(Y = 1 | a_1, z \in \mathcal{Z}'_{h_1,1}) - P(Y = 1 | a_1, z \in \mathcal{Z}'_{h_1,0}) \\
& - P(Y = 1 | a_2, z \in \mathcal{Z}'_{h_2,1}) + P(Y = 1 | a_2, z \in \mathcal{Z}'_{h_2,0}).
\end{aligned}
$$

Similarly, we can define the utility disparity gap under different confidence settings when the decision-maker consistently chooses $T = 0$,

$$
\begin{aligned}
& \Big( \mathbb{E}\left[u(0, Y) | a_1, z \in \mathcal{Z}'_{h_1,1}\right] - \mathbb{E}\left[u(0, Y) | a_1, z \in \mathcal{Z}'_{h_1,0}\right] \Big) \\
& - \Big( \mathbb{E}\left[u(0, Y) | a_2, z \in \mathcal{Z}'_{h_2,1}\right] - \mathbb{E}\left[u(0, Y) | a_2, z \in \mathcal{Z}'_{h_2,0}\right] \Big) \\
& = (u(0, 1) - u(0, 0)) \cdot \Delta_1.
\end{aligned}
\tag{43}
$$

As $P(Y = 1 \mid f_A(z) = a_1, z \in \mathcal{Z}_{h_1,1}) - P(Y = 1 \mid f_A(z) = a_1, z \in \mathcal{Z}_{h_1,0}) \leq 0$, according to Lemma A.2, we have:

$$
\begin{aligned}
& \mathbb{E}\left[u(1, Y) | f_A(z) = a_1, z \in \mathcal{Z}_{h_1,1}\right] - \mathbb{E}\left[u(1, Y) | f_A(z) = a_1, z \in \mathcal{Z}_{h_1,0}\right] \\
& \leq 0 \leq \mathbb{E}\left[u(0, Y) | f_A(z) = a_1, z \in \mathcal{Z}_{h_1,1}\right] - \mathbb{E}\left[u(0, Y) | f_A(z) = a_1, z \in \mathcal{Z}_{h_1,0}\right].
\end{aligned}
\tag{44}
$$

Combining Eqs. 42, 43 and 44, it holds that,

$$
\begin{aligned}
& \Big( \mathbb{E}\left[u(1, Y) | a_2, z \in \mathcal{Z}'_{h_2,1}\right] - \mathbb{E}\left[u(1, Y) | a_2, z \in \mathcal{Z}'_{h_2,0}\right] \Big) \\
& - \Big( \mathbb{E}\left[u(0, Y) | a_2, z \in \mathcal{Z}'_{h_2,1}\right] - \mathbb{E}\left[u(0, Y) | a_2, z \in \mathcal{Z}'_{h_2,0}\right] \Big) \\
& \leq \Big( u(0, 1) - u(0, 0) - u(1, 1) + u(1, 0) \Big) \cdot \Delta_1.
\end{aligned}
\tag{45}
$$

As $P(Y = 1 \mid a_2, z \in \mathcal{Z}_{h_2,1}) - P(Y = 1 \mid a_2, z \in \mathcal{Z}_{h_2,0}) > 0$, it holds that,

$$
\begin{aligned}
& \mathbb{E}\left[u(1, Y) | a_2, z \in \mathcal{Z}_{h_2,1}\right] - \mathbb{E}\left[u(1, Y) | a_2, z \in \mathcal{Z}_{h_2,0}\right] \\
& > 0 \geq \mathbb{E}\left[u(0, Y) | a_2, z \in \mathcal{Z}_{h_2,1}\right] - \mathbb{E}\left[u(0, Y) | a_2, z \in \mathcal{Z}_{h_2,0}\right].
\end{aligned}
\tag{46}
$$

*Based on Eq. 46, the upper bound of the utility disparity of policy $\pi$ is,*

$$
\begin{aligned}
0 \le {} & |\mathbb{E}_\pi\left[u(T,Y)|a_2, z \in \mathcal{Z}_{h_2,1}\right] - \mathbb{E}_\pi\left[u(T,Y)|a_2, z \in \mathcal{Z}_{h_2,0}\right]| \\
\le {} & (\mathbb{E}\left[u(1,Y)|a_2, z \in \mathcal{Z}_{h_2,1}\right] - \mathbb{E}\left[u(1,Y)|a_2, z \in \mathcal{Z}_{h_2,0}\right]) \\
& - (\mathbb{E}\left[u(0,Y)|a_2, z \in \mathcal{Z}_{h_2,1}\right] - \mathbb{E}\left[u(0,Y)|a_2, z \in \mathcal{Z}_{h_2,0}\right]) \\
= {} & \left(1 - \frac{\alpha}{2}\right) \\
& \cdot \Big( \left(\mathbb{E}\left[u(1,Y)|a_2, z \in \mathcal{Z}'_{h_2,1}\right] - \mathbb{E}\left[u(1,Y)|a_2, z \in \mathcal{Z}'_{h_2,0}\right]\right) \\
& - \left(\mathbb{E}\left[u(0,Y)|a_2, z \in \mathcal{Z}'_{h_2,1}\right] - \mathbb{E}\left[u(0,Y)|a_2, z \in \mathcal{Z}'_{h_2,0}\right]\right) \Big) \\
& + \frac{\alpha}{2} \\
& \cdot \Big( \left(\mathbb{E}\left[u(1,Y)|a_2, z \in \mathcal{Z}_{h_2,1} \setminus \mathcal{Z}'_{h_2,1}\right] - \mathbb{E}\left[u(1,Y)|a_2, z \in \mathcal{Z}_{h_2,0} \setminus \mathcal{Z}'_{h_2,0}\right]\right) \\
& - \left(\mathbb{E}\left[u(0,Y)|a_2, z \in \mathcal{Z}_{h_2,1} \setminus \mathcal{Z}'_{h_2,1}\right] - \mathbb{E}\left[u(0,Y)|a_2, z \in \mathcal{Z}_{h_2,0} \setminus \mathcal{Z}'_{h_2,0}\right]\right) \Big).
\end{aligned}
\tag{47}
$$

*For $z \in \mathcal{Z}_{h_2} \setminus \mathcal{Z}'_{h_2}$, as*

$$
\begin{aligned}
u(1,0) &\le \mathbb{E}\left[u(1,Y) \mid a_2, z \in \mathcal{Z}_{h_2} \setminus \mathcal{Z}'_{h_2}\right] \le u(1,1), \\
u(0,1) &\le \mathbb{E}\left[u(0,Y) \mid a_2, z \in \mathcal{Z}_{h_2} \setminus \mathcal{Z}'_{h_2}\right] \le u(0,0).
\end{aligned}
\tag{48}
$$

*we have:*

$$
0 < \mathbb{E}\left[u(1,Y) \mid a_2, z \in \mathcal{Z}_{h_2,1} \setminus \mathcal{Z}'_{h_2,1}\right] - \mathbb{E}\left[u(1,Y) \mid a_2, z \in \mathcal{Z}_{h_2,0} \setminus \mathcal{Z}'_{h_2,0}\right] \le u(1,1) - u(1,0),
\tag{49}
$$

$$
u(0,1) - u(0,0) \le \mathbb{E}\left[u(0,Y) \mid a_2, z \in \mathcal{Z}_{h_2,1} \setminus \mathcal{Z}'_{h_2,1}\right] - \mathbb{E}\left[u(0,Y) \mid a_2, z \in \mathcal{Z}_{h_2,0} \setminus \mathcal{Z}'_{h_2,0}\right] \le 0.
\tag{50}
$$

*Then, Eq. 47 can be reorganized as follows:*

$$
\begin{aligned}
0 \le {} & |\mathbb{E}_\pi\left[u(T,Y)|a_2, z \in \mathcal{Z}_{h_2,1}\right] - \mathbb{E}_\pi\left[u(T,Y)|a_2, z \in \mathcal{Z}_{h_2,0}\right]| \\
\le {} & \left(1 - \frac{\alpha}{2}\right) \cdot \Big(u(1,0) - u(1,1) - u(0,0) + u(0,1)\Big) \cdot \Delta_1 \\
& + \frac{\alpha}{2} \cdot \Big(u(1,1) - u(1,0) - u(0,1) + u(0,0)\Big).
\end{aligned}
\tag{51}
$$

*Based on Eq. 38 and 39, it follows that:*

$$
\Delta_1 \le 0.
\tag{52}
$$

*Therefore, the optimal utility disparity is achieved when:*

$$
\Delta_1 \equiv 0^-.
\tag{53}
$$

*When $f_A$ is perfectly aligned with $f_H$ ($\alpha^* \le \alpha = 0$) and Eq. 53 does not hold, there are infinitely many cases that*

$$
\begin{aligned}
& |\mathbb{E}_\pi\left[u(T,Y)|a_2, z \in \mathcal{Z}_{h_2,1}\right] - \mathbb{E}_\pi\left[u(T,Y)|a_2, z \in \mathcal{Z}_{h_2,0}\right]| \\
& > |\mathbb{E}_{\pi^*}\left[u(T,Y)|a_2, z \in \mathcal{Z}_{h_2,1}\right] - \mathbb{E}_{\pi^*}\left[u(T,Y)|a_2, z \in \mathcal{Z}_{h_2,0}\right]|.
\end{aligned}
\tag{54}
$$

***Case 2.*** *For any confidence $[h_1, a_1]$ with $a_1 > \bar{a}_1$, according to Eq. 37, it holds that,*

$$
P(Y = 1 \mid a_1, z \in \mathcal{Z}_{h_1,1}) - P(Y = 1 \mid a_1, z \in \mathcal{Z}_{h_1,0}) > 0.
\tag{55}
$$

*Furthermore, there exists another $[h_2, a_2]$, where $\bar{a}_2 > a_2 > a_1$ and $h_2 > h_1$ such that,*

$$
P(Y = 1 \mid a_2, z \in \mathcal{Z}_{h_2,1}) - P(Y = 1 \mid a_2, z \in \mathcal{Z}_{h_2,0}) \le 0.
\tag{56}
$$

*According to Lemma A.2, we have:*

$$
\begin{aligned}
& \mathbb{E}\left[u(1,Y)|a_1, z \in \mathcal{Z}_{h_1,1}\right] - \mathbb{E}\left[u(1,Y)|a, z \in \mathcal{Z}_{h,0}\right] \\
& > \mathbb{E}\left[u(0,Y)|a_1, z \in \mathcal{Z}_{h_1,1}\right] - \mathbb{E}\left[u(0,Y)|a, z \in \mathcal{Z}_{h,0}\right].
\end{aligned}
\tag{57}
$$

Combining Eqs. 57, 43 and 44, it holds that,

$$\left( \mathbb{E}\left[ u(0,Y)|a_2, z \in \mathcal{Z}'_{h_2,1} \right] - \mathbb{E}\left[ u(0,Y)|a_2, z \in \mathcal{Z}'_{h_2,0} \right] \right)$$
$$- \left( \mathbb{E}\left[ u(1,Y)|a_2, z \in \mathcal{Z}'_{h_2,1} \right] - \mathbb{E}\left[ u(1,Y)|a_2, z \in \mathcal{Z}'_{h_2,0} \right] \right) \tag{58}$$
$$\leq (u(1,1) - u(1,0) - u(0,1) + u(0,0)) \cdot \Delta_2.$$

where,

$$\Delta_2 = P(Y=1|a_1, z \in \mathcal{Z}'_{h_1,1}) - P(Y=1|a_1, z \in \mathcal{Z}'_{h_1,0})$$
$$- P(Y=1|a_2, z \in \mathcal{Z}'_{h_2,1}) + P(Y=1|a_2, z \in \mathcal{Z}'_{h_2,0})$$

As $P(Y=1 \mid a_2, z \in \mathcal{Z}_{h_2,1}) - P(Y=1 \mid a_2, z \in \mathcal{Z}_{h_2,0}) \leq 0$, it holds that,

$$\mathbb{E}\left[ u(1,Y)|a_2, z \in \mathcal{Z}_{h_2,1} \right] - \mathbb{E}\left[ u(1,Y)|a_2, z \in \mathcal{Z}_{h_2,0} \right]$$
$$\leq 0 \leq \mathbb{E}\left[ u(0,Y)|a_2, z \in \mathcal{Z}_{h_2,1} \right] - \mathbb{E}\left[ u(0,Y)|a_2, z \in \mathcal{Z}_{h_2,0} \right]. \tag{59}$$

Based on Eq. 59, the upper bound of the utility disparity of policy $\pi$ is,

$$0 \leq |\mathbb{E}_\pi\left[ u(T,Y)|a_2, z \in \mathcal{Z}_{h_2,1} \right] - \mathbb{E}_\pi\left[ u(T,Y)|a_2, z \in \mathcal{Z}_{h_2,0} \right]|$$
$$\leq \left( \mathbb{E}\left[ u(0,Y)|a_2, z \in \mathcal{Z}_{h_2,1} \right] - \mathbb{E}\left[ u(0,Y)|a_2, z \in \mathcal{Z}_{h_2,0} \right] \right)$$
$$- \left( \mathbb{E}\left[ u(1,Y)|a_2, z \in \mathcal{Z}_{h_2,1} \right] - \mathbb{E}\left[ u(1,Y)|a_2, z \in \mathcal{Z}_{h_2,0} \right] \right)$$
$$= \left( 1 - \frac{\alpha}{2} \right)$$
$$\cdot \left( \left( \mathbb{E}\left[ u(0,Y)|a_2, z \in \mathcal{Z}'_{h_2,1} \right] - \mathbb{E}\left[ u(0,Y)|a_2, z \in \mathcal{Z}'_{h_2,0} \right] \right) \right. \tag{60}$$
$$\left. - \left( \mathbb{E}\left[ u(1,Y)|a_2, z \in \mathcal{Z}'_{h_2,1} \right] - \mathbb{E}\left[ u(1,Y)|a_2, z \in \mathcal{Z}'_{h_2,0} \right] \right) \right)$$
$$+ \frac{\alpha}{2}$$
$$\cdot \left( \left( \mathbb{E}\left[ u(0,Y)|a_2, z \in \mathcal{Z}_{h_2,1} \setminus \mathcal{Z}'_{h_2,1} \right] - \mathbb{E}\left[ u(0,Y)|a_2, z \in \mathcal{Z}_{h_2,0} \setminus \mathcal{Z}'_{h_2,0} \right] \right) \right.$$
$$\left. - \left( \mathbb{E}\left[ u(1,Y)|a_2, z \in \mathcal{Z}_{h_2,1} \setminus \mathcal{Z}'_{h_2,1} \right] - \mathbb{E}\left[ u(1,Y)|a_2, z \in \mathcal{Z}_{h_2,0} \setminus \mathcal{Z}'_{h_2,0} \right] \right) \right).$$

For $z \in \mathcal{Z}_{h_2} \setminus \mathcal{Z}'_{h_2}$, according to Eq. 48, it holds that,

$$u(1,0) - u(1,1) \leq \mathbb{E}\left[ u(1,Y) \mid a_2, z \in \mathcal{Z}_{h_2,1} \setminus \mathcal{Z}'_{h_2,1} \right] - \mathbb{E}\left[ u(1,Y) \mid a_2, z \in \mathcal{Z}_{h_2,0} \setminus \mathcal{Z}'_{h_2,0} \right] \leq 0, \tag{61}$$

$$0 \leq \mathbb{E}\left[ u(0,Y) \mid a_2, z \in \mathcal{Z}_{h_2,1} \setminus \mathcal{Z}'_{h_2,1} \right] - \mathbb{E}\left[ u(0,Y) \mid a_2, z \in \mathcal{Z}_{h_2,0} \setminus \mathcal{Z}'_{h_2,0} \right] \leq u(0,0) - u(0,1). \tag{62}$$

Then, Eq. 60 can be reorganized as follows:

$$0 \leq |\mathbb{E}_\pi\left[ u(T,Y)|a_2, z \in \mathcal{Z}_{h_2,1} \right] - \mathbb{E}_\pi\left[ u(T,Y)|a_2, z \in \mathcal{Z}_{h_2,0} \right]|$$
$$\leq \left( 1 - \frac{\alpha}{2} \right) \cdot (u(1,1) - u(1,0) - u(0,1) + u(0,0)) \cdot \Delta_2 \tag{63}$$
$$+ \frac{\alpha}{2} \cdot (u(1,1) - u(1,0) - u(0,1) + u(0,0)).$$

Based on Eq. 55 and 56, it follows that:

$$\Delta_2 \geq 0. \tag{64}$$

Therefore, the optimal utility disparity is achieved when:

$$\Delta_2 \equiv 0^+. \tag{65}$$

When $f_A$ is perfectly aligned with $f_H$ ($\alpha^* \leq \alpha = 0$) and Eq. 65 does not hold, there are infinitely many cases that

$$|\mathbb{E}_\pi\left[ u(T,Y)|a_2, z \in \mathcal{Z}_{h_2,1} \right] - \mathbb{E}_\pi\left[ u(T,Y)|a_2, z \in \mathcal{Z}_{h_2,0} \right]|$$
$$> |\mathbb{E}_{\pi^*}\left[ u(T,Y)|a_2, z \in \mathcal{Z}_{h_2,1} \right] - \mathbb{E}_{\pi^*}\left[ u(T,Y)|a_2, z \in \mathcal{Z}_{h_2,0} \right]|. \tag{66}$$

**Case 3.** *For any confidence $[h_1, a_1]$ with $a_1 < \bar{a}_1$, according to Eq. 37, it holds that,*

$$P(Y = 1 \mid a_1, z \in \mathcal{Z}_{h_1,1}) - P(Y = 1 \mid a_1, z \in \mathcal{Z}_{h_1,0}) \leq 0. \tag{67}$$

*Furthermore, there exists another $[h_2, a_2]$, where $\bar{a}_2 > a_2 > a_1$ and $h_2 > h_1$ such that,*

$$P(Y = 1 \mid a_2, z \in \mathcal{Z}_{h_2,1}) - P(Y = 1 \mid a_2, z \in \mathcal{Z}_{h_2,0}) \leq 0. \tag{68}$$

*When the decision-maker consistently chooses $T = 1$, it holds that:*

$$
\begin{aligned}
&\left(\mathbb{E}\left[u(1,Y)|a_1, z \in \mathcal{Z}'_{h_1,1}\right] - \mathbb{E}\left[u(1,Y)|a_1, z \in \mathcal{Z}'_{h_1,0}\right]\right) \\
&+ \left(\mathbb{E}\left[u(1,Y)|a_2, z \in \mathcal{Z}'_{h_2,1}\right] - \mathbb{E}\left[u(1,Y)|a_2, z \in \mathcal{Z}'_{h_2,0}\right]\right) \\
&= (u(1,1) - u(1,0)) \cdot \Delta_3.
\end{aligned}
\tag{69}
$$

*where,*

$$
\begin{aligned}
\Delta_3 = &\, P(Y = 1|a_1, z \in \mathcal{Z}'_{h_1,1}) - P(Y = 1|a_1, z \in \mathcal{Z}'_{h_1,0}) \\
&+ P(Y = 1|a_2, z \in \mathcal{Z}'_{h_2,1}) - P(Y = 1|a_2, z \in \mathcal{Z}'_{h_2,0}).
\end{aligned}
$$

*Similarly, when the decision-maker consistently chooses $T = 0$, it holds that,*

$$
\begin{aligned}
&\left(\mathbb{E}\left[u(0,Y)|a_1, z \in \mathcal{Z}'_{h_1,1}\right] - \mathbb{E}\left[u(0,Y)|a, z \in \mathcal{Z}'_{h_1,0}\right]\right) \\
&+ \left(\mathbb{E}\left[u(0,Y)|a_2, z \in \mathcal{Z}'_{h_2,1}\right] - \mathbb{E}\left[u(0,Y)|a_2, z \in \mathcal{Z}'_{h_2,0}\right]\right) \\
&= (u(0,1) - u(0,0)) \cdot \Delta_3.
\end{aligned}
\tag{70}
$$

*According to Corollary A.2, we have:*

$$
\begin{aligned}
&\mathbb{E}\left[u(1,Y)|a_1, z \in \mathcal{Z}_{h_1,1}\right] - \mathbb{E}\left[u(1,Y)|a_1, z \in \mathcal{Z}_{h_1,0}\right] \\
&\leq \mathbb{E}\left[u(0,Y)|a_1, z \in \mathcal{Z}_{h_1,1}\right] - \mathbb{E}\left[u(0,Y)|a_1, z \in \mathcal{Z}_{h_1,0}\right].
\end{aligned}
\tag{71}
$$

*Combining Eqs. 71, 69 and 70, it holds that,*

$$
\begin{aligned}
&\left(\mathbb{E}\left[u(0,Y)|a_2, z \in \mathcal{Z}'_{h_2,1}\right] - \mathbb{E}\left[u(0,Y)|a_2, z \in \mathcal{Z}'_{h_2,0}\right]\right) \\
&- \left(\mathbb{E}\left[u(1,Y)|a_2, z \in \mathcal{Z}'_{h_2,1}\right] - \mathbb{E}\left[u(1,Y)|a_2, z \in \mathcal{Z}'_{h_2,0}\right]\right) \\
&\leq (u(0,1) - u(0,0) - u(1,1) + u(1,0)) \cdot \Delta_3.
\end{aligned}
\tag{72}
$$

*As $P(Y = 1 \mid a_2, z \in \mathcal{Z}_{h_2,1}) - P(Y = 1 \mid a_2, z \in \mathcal{Z}_{h_2,0}) \leq 0$, it holds that,*

$$
\begin{aligned}
&\mathbb{E}\left[u(1,Y)|a_2, z \in \mathcal{Z}_{h_2,1}\right] - \mathbb{E}\left[u(1,Y)|a_2, z \in \mathcal{Z}_{h_2,0}\right] \\
&\leq 0 \leq \mathbb{E}\left[u(0,Y)|a_2, z \in \mathcal{Z}_{h_2,1}\right] - \mathbb{E}\left[u(0,Y)|a_2, z \in \mathcal{Z}_{h_2,0}\right].
\end{aligned}
\tag{73}
$$

*Based on Eq. 73, the upper bound of the utility disparity of policy $\pi$ is,*

$$
\begin{aligned}
0 \leq &\left|\mathbb{E}_\pi\left[u(T,Y)|a_2, z \in \mathcal{Z}_{h_2,1}\right] - \mathbb{E}_\pi\left[u(T,Y)|a_2, z \in \mathcal{Z}_{h_2,0}\right]\right| \\
\leq &\left(\mathbb{E}\left[u(0,Y)|a_2, z \in \mathcal{Z}_{h_2,1}\right] - \mathbb{E}\left[u(0,Y)|a_2, z \in \mathcal{Z}_{h_2,0}\right]\right) \\
&- \left(\mathbb{E}\left[u(1,Y)|a_2, z \in \mathcal{Z}_{h_2,1}\right] - \mathbb{E}\left[u(1,Y)|a_2, z \in \mathcal{Z}_{h_2,0}\right]\right) \\
= &\left(1 - \frac{\alpha}{2}\right) \\
&\cdot \left(\left(\mathbb{E}\left[u(0,Y)|a_2, z \in \mathcal{Z}'_{h_2,1}\right] - \mathbb{E}\left[u(0,Y)|a_2, z \in \mathcal{Z}'_{h_2,0}\right]\right)\right. \\
&- \left.\left(\mathbb{E}\left[u(1,Y)|a_2, z \in \mathcal{Z}'_{h_2,1}\right] - \mathbb{E}\left[u(1,Y)|a_2, z \in \mathcal{Z}'_{h_2,0}\right]\right)\right) \\
&+ \frac{\alpha}{2} \\
&\cdot \left(\left(\mathbb{E}\left[u(0,Y)|a_2, z \in \mathcal{Z}_{h_2,1} \setminus \mathcal{Z}'_{h_2,1}\right] - \mathbb{E}\left[u(0,Y)|a_2, z \in \mathcal{Z}_{h_2,0} \setminus \mathcal{Z}'_{h_2,0}\right]\right)\right. \\
&- \left.\left(\mathbb{E}\left[u(1,Y)|a_2, z \in \mathcal{Z}_{h_2,1} \setminus \mathcal{Z}'_{h_2,1}\right] - \mathbb{E}\left[u(1,Y)|a_2, z \in \mathcal{Z}_{h_2,0} \setminus \mathcal{Z}'_{h_2,0}\right]\right)\right).
\end{aligned}
\tag{74}
$$

*For $z \in \mathcal{Z}_{h_2} \setminus \mathcal{Z}'_{h_2}$, according to Eq. 48, it holds that,*

$$u(1,0) - u(1,1) \leq \mathbb{E}\left[u(1,Y) \mid a_2, z \in \mathcal{Z}_{h_2,1} \setminus \mathcal{Z}'_{h_2,1}\right] - \mathbb{E}\left[u(1,Y) \mid a_2, z \in \mathcal{Z}_{h_2,0} \setminus \mathcal{Z}'_{h_2,0}\right] \leq 0. \tag{75}$$

$$0 \leq \mathbb{E}\left[u(0,Y) \mid a_2, z \in \mathcal{Z}_{h_2,1} \setminus \mathcal{Z}'_{h_2,1}\right] - \mathbb{E}\left[u(0,Y) \mid a_2, z \in \mathcal{Z}_{h_2,0} \setminus \mathcal{Z}'_{h_2,0}\right] \leq u(0,0) - u(0,1). \tag{76}$$

*Then, Eq. 74 can be reorganized as follows:*

$$0 \leq |\mathbb{E}_\pi\left[u(T,Y)|a_2, z \in \mathcal{Z}_{h_2,1}\right] - \mathbb{E}_\pi\left[u(T,Y)|a_2, z \in \mathcal{Z}_{h_2,0}\right]|$$
$$\leq \left(1 - \frac{\alpha}{2}\right) \cdot (u(1,0) - u(1,1) - u(0,0) + u(0,1)) \cdot \Delta_3 \tag{77}$$
$$+ \frac{\alpha}{2} \cdot (u(1,1) - u(1,0) - u(0,1) + u(0,0)).$$

*Based on Eq. 67 and 68, it follows that:*

$$\Delta_3 \leq 0. \tag{78}$$

*Therefore, the optimal utility disparity is achieved when:*

$$\Delta_3 \equiv 0^-. \tag{79}$$

*When $f_A$ is perfectly aligned with $f_H$ ($\alpha^* \leq \alpha = 0$) and Eq. 79 does not hold, there are infinitely many cases that*

$$|\mathbb{E}_\pi\left[u(T,Y)|a_2, z \in \mathcal{Z}_{h_2,1}\right] - \mathbb{E}_\pi\left[u(T,Y)|a_2, z \in \mathcal{Z}_{h_2,0}\right]|$$
$$> |\mathbb{E}_{\pi^*}\left[u(T,Y)|a_2, z \in \mathcal{Z}_{h_2,1}\right] - \mathbb{E}_{\pi^*}\left[u(T,Y)|a_2, z \in \mathcal{Z}_{h_2,0}\right]|. \tag{80}$$

**Case 4.** *For any confidence $[h_1, a_1]$ with $a_1 > \bar{a}_1$, according to Eq. 37, it holds that,*

$$P(Y = 1 \mid a_1, z \in \mathcal{Z}_{h_1,1}) - P(Y = 1 \mid a_1, z \in \mathcal{Z}_{h_1,0}) > 0. \tag{81}$$

*Furthermore, there exists another $[h_2, a_2]$, where $a_2 > a_1 > \bar{a}_2$ and $h_2 > h_1$ such that,*

$$P(Y = 1 \mid a_2, z \in \mathcal{Z}_{h_2,1}) - P(Y = 1 \mid a_2, z \in \mathcal{Z}_{h_2,0}) > 0. \tag{82}$$

*According to Lemma A.2, we have:*

$$\mathbb{E}\left[u(1,Y)|a_1, z \in \mathcal{Z}_{h_1,1}\right] - \mathbb{E}\left[u(1,Y)|a_1, z \in \mathcal{Z}_{h_1,0}\right]$$
$$> \mathbb{E}\left[u(0,Y)|a_1, z \in \mathcal{Z}_{h_1,1}\right] - \mathbb{E}\left[u(0,Y)|a_1, z \in \mathcal{Z}_{h_1,0}\right]. \tag{83}$$

*Combining Eqs. 83, 69 and 70, it holds that,*

$$\left(\mathbb{E}\left[u(1,Y)|a_2, z \in \mathcal{Z}'_{h_2,1}\right] - \mathbb{E}\left[u(1,Y)|a_2, z \in \mathcal{Z}'_{h_2,0}\right]\right)$$
$$- \left(\mathbb{E}\left[u(0,Y)|a_2, z \in \mathcal{Z}'_{h_2,1}\right] - \mathbb{E}\left[u(0,Y)|a_2, z \in \mathcal{Z}'_{h_2,0}\right]\right) \tag{84}$$
$$\leq (u(1,1) - u(1,0) - u(0,1) + u(0,0)) \cdot \Delta_4.$$

*where,*

$$\Delta_4 = P(Y = 1|a_1, z \in \mathcal{Z}'_{h_1,1}) - P(Y = 1|a_1, z \in \mathcal{Z}'_{h_1,0})$$
$$+ P(Y = 1|a_2, z \in \mathcal{Z}'_{h_2,1}) - P(Y = 1|a_2, z \in \mathcal{Z}'_{h_2,0}).$$

*As $P(Y = 1 \mid a_2, z \in \mathcal{Z}_{h_2,1}) - P(Y = 1 \mid a_2, z \in \mathcal{Z}_{h_2,0}) > 0$, it holds that,*

$$\mathbb{E}\left[u(0,Y)|a_2, z \in \mathcal{Z}_{h_2,1}\right] - \mathbb{E}\left[u(0,Y)|a_2, z \in \mathcal{Z}_{h_2,0}\right]$$
$$\leq 0 < \mathbb{E}\left[u(1,Y)|a_2, z \in \mathcal{Z}_{h_2,1}\right] - \mathbb{E}\left[u(1,Y)|a_2, z \in \mathcal{Z}_{h_2,0}\right]. \tag{85}$$

*Based on Eq. 85, the upper bound of the utility disparity of policy $\pi$ is,*

$$0 \leq |\mathbb{E}_\pi\left[u(T,Y)|a_2, z \in \mathcal{Z}_{h_2,1}\right] - \mathbb{E}_\pi\left[u(T,Y)|a_2, z \in \mathcal{Z}_{h_2,0}\right]|$$
$$\leq \left(\mathbb{E}\left[u(1,Y)|a_2, z \in \mathcal{Z}_{h_2,1}\right] - \mathbb{E}\left[u(1,Y)|a_2, z \in \mathcal{Z}_{h_2,0}\right]\right)$$
$$- \left(\mathbb{E}\left[u(0,Y)|a_2, z \in \mathcal{Z}_{h_2,1}\right] - \mathbb{E}\left[u(0,Y)|a_2, z \in \mathcal{Z}_{h_2,0}\right]\right)$$
$$= \left(1 - \frac{\alpha}{2}\right)$$
$$\cdot \left(\left(\mathbb{E}\left[u(1,Y)|a_2, z \in \mathcal{Z}'_{h_2,1}\right] - \mathbb{E}\left[u(1,Y)|a_2, z \in \mathcal{Z}'_{h_2,0}\right]\right)\right.$$
$$\left. - \left(\mathbb{E}\left[u(0,Y)|a_2, z \in \mathcal{Z}'_{h_2,1}\right] - \mathbb{E}\left[u(0,Y)|a_2, z \in \mathcal{Z}'_{h_2,0}\right]\right)\right) \tag{86}$$
$$+ \frac{\alpha}{2}$$
$$\cdot \left(\left(\mathbb{E}\left[u(1,Y)|a_2, z \in \mathcal{Z}_{h_2,1} \setminus \mathcal{Z}'_{h_2,1}\right] - \mathbb{E}\left[u(1,Y)|a_2, z \in \mathcal{Z}_{h_2,0} \setminus \mathcal{Z}'_{h_2,0}\right]\right)\right.$$
$$\left. - \left(\mathbb{E}\left[u(0,Y)|a_2, z \in \mathcal{Z}_{h_2,1} \setminus \mathcal{Z}'_{h_2,1}\right] - \mathbb{E}\left[u(0,Y)|a_2, z \in \mathcal{Z}_{h_2,0} \setminus \mathcal{Z}'_{h_2,0}\right]\right)\right).$$

*For $z \in \mathcal{Z}_{h_2} \setminus \mathcal{Z}'_{h_2}$, according to Eq. 48, it holds that,*

$$0 \leq \mathbb{E}\left[u(1,Y) \mid a_2, z \in \mathcal{Z}_{h_2,1} \setminus \mathcal{Z}'_{h_2,1}\right] - \mathbb{E}\left[u(1,Y) \mid a_2, z \in \mathcal{Z}_{h_2,0} \setminus \mathcal{Z}'_{h_2,0}\right] \leq u(1,1) - u(1,0), \tag{87}$$

$$u(0,1) - u(0,0) \leq \mathbb{E}\left[u(0,Y) \mid a_2, z \in \mathcal{Z}_{h_2,1} \setminus \mathcal{Z}'_{h_2,1}\right] - \mathbb{E}\left[u(0,Y) \mid a_2, z \in \mathcal{Z}_{h_2,0} \setminus \mathcal{Z}'_{h_2,0}\right] \leq 0. \tag{88}$$

*Then, Eq. 86 can be reorganized as follows:*

$$0 \leq \left|\mathbb{E}_\pi\left[u(T,Y)|a_2, z \in \mathcal{Z}_{h_2,1}\right] - \mathbb{E}_\pi\left[u(T,Y)|a_2, z \in \mathcal{Z}_{h_2,0}\right]\right|$$
$$\leq \left(1 - \frac{\alpha}{2}\right) \cdot (u(1,1) - u(1,0) - u(0,1) + u(0,0)) \cdot \Delta_4 \tag{89}$$
$$+ \frac{\alpha}{2} \cdot (u(1,1) - u(1,0) - u(0,1) + u(0,0)).$$

*Based on Eq. 81 and 82, it follows that:*

$$\Delta_4 > 0. \tag{90}$$

*Therefore, the optimal utility disparity is achieved when:*

$$\Delta_4 \equiv 0^+. \tag{91}$$

*When $f_A$ is perfectly aligned with $f_H$ ($\alpha^* \leq \alpha = 0$) and Eq. 91 does not hold, there are infinitely many cases that*

$$\left|\mathbb{E}_\pi\left[u(T,Y)|a_2, z \in \mathcal{Z}'_{h_2,1}\right] - \mathbb{E}_\pi\left[u(T,Y)|a_2, z \in \mathcal{Z}'_{h_2,0}\right]\right|$$
$$> \left|\mathbb{E}_{\pi^*}\left[u(T,Y)|a_2, z \in \mathcal{Z}'_{h_2,1}\right] - \mathbb{E}_{\pi^*}\left[u(T,Y)|a_2, z \in \mathcal{Z}'_{h_2,0}\right]\right|. \tag{92}$$

*Based on the above proof, we have demonstrated the existence of scenarios in which, even when $f_A$ is perfectly aligned with $f_H$, any monotone policy $\pi$ leads to a suboptimal utility disparity, thereby supporting Theorem 3.4.* $\qquad\square$

## A.4 Proof of Theorem 3.6

**Theorem 3.6** (Utility disparity upper bound under $\alpha$-human-alignment) For a given AI-assisted decision-making process with a utility function $u(T,Y)$ satisfying Eq. 1 and the human decision-maker with any monotone AI-assisted decision policy $\pi \in \Pi(H, A)$, if the AI confidence function $f_A$ is $\alpha_h$-human-alignment and satisfies $\alpha_g$-inter-group-alignment, then the utility disparity is bounded by,

$$\left|\mathbb{E}_\pi\left[u(T,Y)|f_A(z) = a, z \in \mathcal{Z}_{h,1}\right] - \mathbb{E}_\pi\left[u(T,Y)|f_A(z) = a, z \in \mathcal{Z}_{h,0}\right]\right|$$
$$\leq (u(1,1) - u(0,1) - u(1,0) + u(0,0)) \cdot \left(\frac{\alpha_h}{2} + \left(1 - \frac{\alpha_h}{2}\right) \cdot \left(3\alpha_g - \alpha_g^2\right)\right). \tag{93}$$

***Proof.*** *Given $\alpha_g$-inter-group-alignment, for any two confidence levels $\{h_1, a_1\}$ and $\{h_2, a_2\}$ with $a_2 > a_1$ and $h_2 > h_1$, the following conditions hold for all $z \in \mathcal{Z}''_{h_1}$ and $z \in \mathcal{Z}''_{h_2}$, respectively:*

$$-\alpha_g \leq P(Y = 1|a_1, z \in \mathcal{Z}''_{h_1,1}) - P(Y = 1|a_1, z \in \mathcal{Z}''_{h_1,0}) \leq \alpha_g. \tag{94}$$

$$-\alpha_g \leq P(Y = 1|a_2, z \in \mathcal{Z}''_{h_2,1}) - P(Y = 1|a_2, z \in \mathcal{Z}''_{h_2,0}) \leq \alpha_g, \tag{95}$$

*where $\mathcal{Z}''_{h_1} \subset \mathcal{Z}_{h_1}$ and $\mathcal{Z}''_{h_2} \subset \mathcal{Z}_{h_2}$ with $\left|\mathcal{Z}''_{h_1}\right| \geq (1 - \alpha_g/2) \cdot |\mathcal{Z}_h|$ and $\left|\mathcal{Z}''_{h_2}\right| \geq (1 - \alpha_g/2) \cdot |\mathcal{Z}_{h_2}|$. Referring to Case 1 in Appendix A.3, we have the following utility disparity upper bound for AI confidence levels $\{h_2, a_2\}$:*

$$0 \leq \left|\mathbb{E}_\pi\left[u(T,Y)|a_2, z \in \mathcal{Z}_{h_2,1}\right] - \mathbb{E}_\pi\left[u(T,Y)|a_2, z \in \mathcal{Z}_{h_2,0}\right]\right|$$
$$\leq \left(1 - \frac{\alpha}{2}\right) \cdot (u(1,0) - u(1,1) - u(0,0) + u(0,1)) \cdot \Delta_1 \tag{96}$$
$$+ \frac{\alpha}{2} \cdot (u(1,1) - u(1,0) - u(0,1) + u(0,0)).$$

*where,*

$$\Delta_1 = P(Y = 1|a_1, z \in \mathcal{Z}'_{h_1,1}) - P(Y = 1|a_1, z \in \mathcal{Z}'_{h_1,0})$$
$$- P(Y = 1|a_2, z \in \mathcal{Z}'_{h_2,1}) + P(Y = 1|a_2, z \in \mathcal{Z}'_{h_2,0}).$$

*Using the conditions given by Eqs. 94 and 95, we have:*

$$-2\alpha_g \leq \Delta_2 \leq 2\alpha_g. \tag{97}$$

*where,*

$$\Delta_2 = P(Y = 1|a_1, z \in \mathcal{Z}'_{h_1,1} \cap \mathcal{Z}''_{h_1,1}) - P(Y = 1|a_1, z \in \mathcal{Z}'_{h_1,0} \cap \mathcal{Z}''_{h_1,0})$$
$$- P(Y = 1|a_2, z \in \mathcal{Z}'_{h_2,1} \cap \mathcal{Z}''_{h_2,1}) + P(Y = 1|a_2, z \in \mathcal{Z}'_{h_2,0} \cap \mathcal{Z}''_{h_2,0}).$$

*For $z \in \mathcal{Z}'_h \setminus \mathcal{Z}''_h$, it holds that,*

$$-2 \leq \Delta_3 \leq 2. \tag{98}$$

*where,*

$$\Delta_3 = P(Y = 1|a_1, z \in \mathcal{Z}'_{h_1,1} \setminus \mathcal{Z}''_{h_1,1}) - P(Y = 1|a_1, z \in \mathcal{Z}'_{h_1,0} \setminus \mathcal{Z}''_{h_1,0})$$
$$- P(Y = 1|a_2, z \in \mathcal{Z}'_{h_2,1} \setminus \mathcal{Z}''_{h_2,1}) + P(Y = 1|a_2, z \in \mathcal{Z}'_{h_2,0} \setminus \mathcal{Z}''_{h_2,0}).$$

*Incorporating these conditions into the utility disparity upper bound in Eq. 96, we get:*

$$0 \leq \left| \mathbb{E}_\pi \left[ u(T, Y)|a_2, z \in \mathcal{Z}_{h_2,1} \right] - \mathbb{E}_\pi \left[ u(T, Y)|a_2, z \in \mathcal{Z}_{h_2,0} \right] \right|$$
$$\leq (u(1,1) - u(1,0) - u(0,1) + u(0,0)) \cdot \left( \left( 1 - \frac{\alpha_h}{2} \right) \cdot \left( \left( 1 - \frac{\alpha_g}{2} \right) \cdot 2\alpha_g + \frac{\alpha_g}{2} \cdot 2 \right) + \frac{\alpha_h}{2} \right)$$
$$= (u(1,1) - u(0,1) - u(1,0) + u(0,0)) \cdot \left( \frac{\alpha_h}{2} + \left( 1 - \frac{\alpha_h}{2} \right) \cdot \left( 3\alpha_g - \alpha_g^2 \right) \right). \tag{99}$$

*This bound can be similarly derived for Cases 2~4 in Appendix A.3, yielding a consistent utility disparity bound as stated in Theorem 3.6.* □

## A.5 PROOF OF THEOREM 4.4

**Theorem 4.4** ($\alpha/2$-Group-level multicalibration leads to $\alpha$-human-alignment and $\alpha$-inter-group-alignment meanwhile) Let $f_A : \mathcal{Z} \rightarrow [0,1]$ be an AI's confidence function. Suppose for each human decision-maker group $i \in \{1, ..., |S|\}$, $f_A(z|z \in \mathcal{Z}_{h,s_i})$ is $\alpha/2$-multicalibrated with respect to the collection $\mathcal{C} = \{\mathcal{Z}_{h,s_i}\}_{h \in \mathcal{H}}$ with $\mathcal{Z}_{h,s_i} = \{(x, h, s) | f_H(x) = h, S = s_i\}$, then $f_A$ is both $\alpha$-aligned with respect to the human decision-maker's confidence function $f_H$ and $\alpha$-inter-group aligned across the human decision-maker groups.

**Proof.** *If in the $i$-th sensitive group, $f_A$ is $\alpha/2$-multicalibration with respect to $\{\mathcal{Z}_{h,s_i}\}_{h \in \mathcal{H}}$, then, according to the Definition 3.1, for any $h \in \mathcal{H}$, there exists $\mathcal{Z}'_h \subset \mathcal{Z}_h$ with $|\mathcal{Z}'_h| \geq \left( 1 - \frac{\alpha}{2} \right) \cdot |\mathcal{Z}_h|$ such that, for any AI confidence $a_1, h_1 \in [0, 1]$, it holds that,*

$$\left| P(Y = 1 \mid f_A(z) = a_1, z \in \mathcal{Z}'_{h_1,1}) - a_1 \right| \leq \frac{\alpha}{2}. \tag{100}$$

$$\left| P(Y = 1 \mid f_A(z) = a_1, z \in \mathcal{Z}'_{h_1,0}) - a_1 \right| \leq \frac{\alpha}{2}. \tag{101}$$

*From the given inequalities, we have:*

$$P(Y = 1 \mid f_A(z) = a_1, z \in \mathcal{Z}'_{h_1,1}) \in \left[ a_1 - \frac{\alpha}{2}, a_1 + \frac{\alpha}{2} \right], \tag{102}$$

$$P(Y = 1 \mid f_A(z) = a_1, z \in \mathcal{Z}'_{h_1,0}) \in \left[ a_1 - \frac{\alpha}{2}, a_1 + \frac{\alpha}{2} \right]. \tag{103}$$

*Then, it's natural that $f_A$ satisfied $\alpha$-inter-group-alignment as,*

$$\left| P(Y = 1 \mid f_A(z) = a_1, z \in \mathcal{Z}'_{h_1,1}) - P(Y = 1 \mid f_A(z) = a_1, z \in \mathcal{Z}'_{h_1,0}) \right| \leq \alpha. \tag{104}$$

*As for any human decision-maker confidence $0 \leq h_1 \leq h_2 \leq 1$ and $0 \leq a_1 \leq a_2 \leq 1$, it holds that,*

$$P(Y = 1 \mid f_A(z) = a_2, z \in \mathcal{Z}'_{h_2,1}) \in \left[ a_2 - \frac{\alpha}{2}, a_2 + \frac{\alpha}{2} \right]. \tag{105}$$

$$P(Y = 1 \mid f_A(z) = a_2, z \in \mathcal{Z}'_{h_2,0}) \in \left[ a_2 - \frac{\alpha}{2}, a_2 + \frac{\alpha}{2} \right]. \tag{106}$$

*For $P(S = 1|a, z \in \mathcal{Z}'_h) + P(S = 0|a, z \in \mathcal{Z}'_h) = 1$, it holds that,*

$$
\begin{aligned}
& P(Y = 1|a_1, z \in \mathcal{Z}'_{h_1}) - P(Y = 1|a_2, z \in \mathcal{Z}'_{h_2}) \\
&= P(Y = 1|a_1, z \in \mathcal{Z}'_{h_1,1}) \cdot P(S = 1|a_1, z \in \mathcal{Z}'_{h_1}) \\
&\quad + P(Y = 1|a_1, z \in \mathcal{Z}'_{h_1,0}) \cdot P(S = 0|a_1, z \in \mathcal{Z}'_{h_1}) \\
&\quad - P(Y = 1|a_2, z \in \mathcal{Z}'_{h_2,1}) \cdot P(S = 1|a_2, z \in \mathcal{Z}'_{h_2}) \\
&\quad - P(Y = 1|a_2, z \in \mathcal{Z}'_{h_2,0}) \cdot P(S = 0|a_2, z \in \mathcal{Z}'_{h_2}) \\
&= P(Y = 1|a_1, z \in \mathcal{Z}'_{h_1,1}) \cdot P(S = 1|a_1, z \in \mathcal{Z}'_{h_1}) \\
&\quad + P(Y = 1|a_1, z \in \mathcal{Z}'_{h_1,0}) \cdot \left(1 - P(S = 1|a_1, z \in \mathcal{Z}'_{h_1})\right) \\
&\quad - P(Y = 1|a_2, z \in \mathcal{Z}'_{h_2,1}) \cdot P(S = 1|a_2, z \in \mathcal{Z}'_{h_2}) \\
&\quad - P(Y = 1|a_2, z \in \mathcal{Z}'_{h_2,0}) \cdot \left(1 - P(S = 1|a_2, z \in \mathcal{Z}'_{h_2})\right) \\
&\leq P(S = 1|a_1, z \in \mathcal{Z}'_{h_1}) \cdot (a_1 + \frac{\alpha}{2}) + \left(1 - P(S = 1|a_1, z \in \mathcal{Z}'_{h_1})\right) \cdot (a_1 + \frac{\alpha}{2}) \\
&\quad - P(S = 1|a_2, z \in \mathcal{Z}'_{h_2}) \cdot (a_2 - \frac{2}{\alpha}) - \left(1 - P(S = 1|a_2, z \in \mathcal{Z}'_{h_2})\right) \cdot (a_2 - \frac{2}{\alpha}) \\
&= \alpha + a_1 - a_2 \leq \alpha.
\end{aligned}
\tag{107}
$$

*As $f_A$ is $\alpha/2$-multicalibrated with respect to the collection $\mathcal{C}$, this implies that $f_A$ is $\alpha/2$-calibrated with respect to any of the sets $\mathcal{Z}_h \in \mathcal{Z}$. Consequently, $f_A$ satisfies $\alpha$-human-alignment and $\alpha$-inner-group-alignment meanwhile.* □

### A.6 ALGORITHM OF GROUP-LEVEL CONFIDENCE MULTICALIBRATION

The procedure is outlined in Algorithm 1.

### A.7 EXPERIMENT SETTINGS

#### A.7.1 DATASET PROCESSING

Following the data processing (Corvelo Benz & Rodriguez, 2023), we transform the original dataset's confidence values from a scale of $[-1, 1]$ to $[0, 1]$ to ensure consistency with our human-AI interactive model (Section 2). In the original dataset, predictions by participants are from different but overlapping sets of countries across tasks, who are told the AI advice has different accuracy. Thus, to control for these confounding factors, we focus exclusively on participants from the United States who are informed that the AI advice had an $80\%$ accuracy. Furthermore, we use gender as a sensitive attribute, a recommended factor that may influence expertise of human decision-makers across different tasks but should be treated with equal utility in AI assistance for social good (Zappalà et al., 2024; Ward et al., 2022). We exclude records where gender information is not provided. The data are then preprocessed to filter out samples with missing information and confounding factors (Appendix A.7.1), resulting in $14,999$ AI-assisted decision-making records from $469$ participants overall, as detailed in Table 2.

#### A.7.2 DISCRETIZATION PROCESS

In the following, we present a detailed description of the discretization parameters used in our experiments: the human confidence $h$ is discretized into 3 bins per task, $\{\mathcal{H}_1, \mathcal{H}_2, \mathcal{H}_3\}$, corresponding to low, medium, and high confidence levels, respectively. The bin boundaries are set such that each bin contains approximately equal probability mass, with the bin values assigned as the average confidence within each bin. The AI's confidence $a$ are divided into uniformly sized bins per task with centred value given by $\Lambda = \{\frac{\lambda}{2}, \frac{3\lambda}{2}, \dots, 1 - \frac{\lambda}{2}\}$, where $\lambda = 1/8$. The above process discretizes the continuous confidence space $\mathcal{H} \times \mathcal{A}$ into a grid of $3 \times \lfloor 1/\lambda \rfloor$ cells. The $(i, j)$-th grid cell contains the samples $\mathcal{Z}^j_i = \{(x, h, s) \in \mathcal{Z}|h \in \mathcal{H}_i, f_A(z) \in [\Lambda[j] - \lambda/2, \Lambda[j] + \lambda/2)\}$. Furthermore, based on the value of the sensitive attribute $s \in \mathcal{S}$, the samples in $\mathcal{Z}^j_i$ can be further divided into $\mathcal{Z}^j_i =$

---

**Algorithm 1:** Group-Level Confidence Multicalibration

---

**Input:** $\widetilde{\alpha}, \lambda$

**Result:** calibrated AI confidence function: $f_A$

1 Initialize $\mathcal{C}_1 \leftarrow \{\mathcal{Z}_{h,1}\}_{h \in \mathcal{H}}$ with $\mathcal{Z}_{h,1} \leftarrow \{(x,h,s) \mid f_H(x) = h, S = 1\}$

2 Initialize $\mathcal{C}_0 \leftarrow \{\mathcal{Z}_{h,0}\}_{h \in \mathcal{H}}$ with $\mathcal{Z}_{h,0} \leftarrow \{(x,h,s) \mid f_H(x) = h, S = 0\}$

3 **repeat**

4    **for** $\mathcal{Z}_{h,1} \in \mathcal{C}_1$ **do**

5       **for** $a \in \{1, ..., \lfloor 1/\lambda \rfloor\}$ **do**

6          Let $\mathcal{Z}_{h,1}^a \rightarrow \mathcal{Z}_{h,1} \cap \{z \mid f_A(z) \in [\Lambda[a] - \lambda/2, \Lambda[a] + \lambda/2)\}$

7          **if** $P(z \in \mathcal{Z}_{h,1}^a) < \widetilde{\alpha}\lambda \cdot P(z \in \mathcal{Z}_{h,1})$ **then**

8             **continue**

9          **end**

10          **if** $\left| E\left[f_A(z) \mid z \in \mathcal{Z}_{h,1}^a\right] - P(Y = 1 \mid z \in \mathcal{Z}_{h,1}^a) \right| \leq \widetilde{\alpha}$ **then**

11             $f_A(z) \rightarrow f_A(z) + P(Y = 1 \mid z \in \mathcal{Z}_{h,1}^a) - \mathbb{E}\left[f_A(z) \mid z \in \mathcal{Z}_{h,1}^a\right]$ for all $z \in \mathcal{Z}_{h,1}^a$

12          **end**

13       **end**

14    **end**

15 **until** *no* $\mathcal{Z}_{h,1}^a$ *updated*;

16 **for** $a \in \{1, ..., \lfloor 1/\lambda \rfloor\}$ **do**

17    $f_A(z) \rightarrow \mathbb{E}\left[f_A(z) \mid \cup_{h \in \mathcal{H}} \mathcal{C}_1 \cap \{z \mid f_A(z) \in [\Lambda[a] - \lambda/2, \Lambda[a] + \lambda/2)\}\right]$

18 **end**

19 **repeat**

20    **for** $\mathcal{Z}_{h,0} \in \mathcal{C}_0$ **do**

21       **for** $a \in \{1, ..., \lfloor 1/\lambda \rfloor\}$ **do**

22          Let $\mathcal{Z}_{h,0}^a \rightarrow \mathcal{Z}_{h,0} \cap \{z \mid f_A(z) \in [\Lambda[a] - \lambda/2, \Lambda[a] + \lambda/2)\}$

23          **if** $P(z \in \mathcal{Z}_{h,0}^a) < \widetilde{\alpha}\lambda \cdot P(z \in \mathcal{Z}_{h,0})$ **then**

24             **continue**

25          **end**

26          **if** $\left| E\left[f_A(z) \mid z \in \mathcal{Z}_{h,0}^a\right] - P(Y = 1 \mid z \in \mathcal{Z}_{h,0}^a) \right| \leq \widetilde{\alpha}$ **then**

27             $f_A(z) \rightarrow f_A(z) + P(Y = 1 \mid z \in \mathcal{Z}_{h,0}^a) - \mathbb{E}\left[f_A(z) \mid z \in \mathcal{Z}_{h,0}^a\right]$ for all $z \in \mathcal{Z}_{h,0}^a$

28          **end**

29       **end**

30    **end**

31 **until** *no* $\mathcal{Z}_{h,0}^a$ *updated*;

32 **for** $a \in \{1, ..., \lfloor 1/\lambda \rfloor\}$ **do**

33    $f_A(z) \rightarrow \mathbb{E}\left[f_A(z) \mid \cup_{h \in \mathcal{H}} \mathcal{C}_0 \cap \{z \mid f_A(z) \in [\Lambda[a] - \lambda/2, \Lambda[a] + \lambda/2)\}\right]$

34 **end**

35 **return** $f_A$

---

Table 2: The details of human-AI interactions dataset (grouped by "gender")

| Experiment | Task | Type | Decision Record | Human Decision-makers Count | |
|---|---|---|---|---|---|
| | | | | S=0 | S=1 |
| 1 | Art | image | 4637 | 77 | 68 |
| 2 | Cities | image | 2878 | 52 | 38 |
| 3 | Sarcasm | text | 4543 | 70 | 72 |
| 4 | Census | tabular | 2941 | 49 | 43 |

$\left\{\mathcal{Z}_{i,1}^j, \mathcal{Z}_{i,0}^j\right\}$, where $\mathcal{Z}_{i,1}^j = \{(x,h,s) \in \mathcal{Z} \mid h \in \mathcal{H}_i, f_A(z) \in [\Lambda[j] - \lambda/2, \Lambda[j] + \lambda/2), S = 1\}$ and $\mathcal{Z}_{i,0}^j = \{(x,h,s) \in \mathcal{Z} \mid h \in \mathcal{H}_i, f_A(z) \in [\Lambda[j] - \lambda/2, \Lambda[j] + \lambda/2), S = 0\}$.

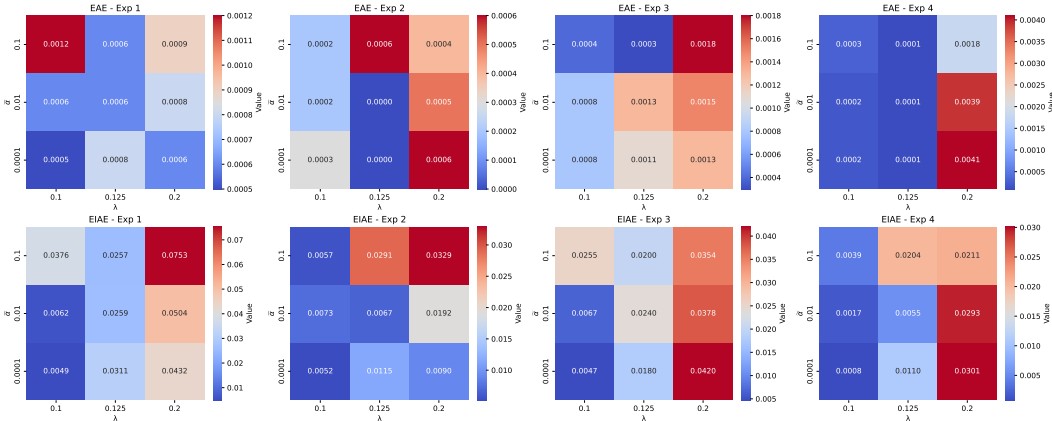

Figure 4: Visualization of EAE and EIAE metrics following group-level multicalibration, with parameters $\widetilde{\alpha} \sim [0.0001, 0.01, 0.1]$ and $\lambda \sim [0.1, 0.125, 0.2]$.

Table 3: The details of human-AI interactions dataset (grouped by "education")

| Experiment | Task | Type | Decision Record | Human Decision-makers Count | |
|---|---|---|---|---|---|
| | | | | S=0 | S=1 |
| **1** | **Art** | image | 4637 | 27 | 118 |
| **2** | **Cities** | image | 2878 | 13 | 77 |
| **3** | **Sarcasm** | text | 4543 | 23 | 119 |
| **4** | **Census** | tabular | 2941 | 14 | 78 |

## A.8 ADDITIONAL EXPERIMENTS

### A.8.1 IMPACT OF DISCRETIZATION PARAMETERS ON GROUP-LEVEL MULTICALIBRATION EFFICIENCY

Figure 4 presents the evaluation of EAE and EIAE metrics across 4 tasks after performing group-level calibration with varying hyperparameters $\widetilde{\alpha} \sim [0.0001, 0.01, 0.1]$ and $\lambda \sim [0.1, 0.125, 0.2]$. The results indicate a general trend where smaller values of $\widetilde{\alpha} + \lambda$ (reflected in results closer to the left or bottom of the plots) lead to lower EAE and EIAE values. We focus on the general trend as we have claimed that discrete evaluation metrics like EAE and EIAE provide a more accurate reflection of alignment when there are significant changes and may fail to capture alignment across the entire continuous confidence space fully. Nevertheless, the general trend strongly supports the notion that decreasing $\widetilde{\alpha} + \lambda$ leads to improved human-alignment and inter-group-alignment.

### A.8.2 GENERALIZATION TO MULTIPLE GROUPS

In this experiment, we introduce the additional demographic feature, "education," which contains numerical values ranging in $2 \sim 8$, reflecting human decision-makers with varying levels of educational attainment. We transform this feature into a binary demographic variable: decision-makers with "education"$> 6$ are categorized as Group $S = 0$, while those with "education"$\leq 6$ are categorized as Group $S = 1$.

We first validate the effectiveness of our method under the new demographic feature "education." The count of human decision-markers across different groups is shown in Table 3. The experimental results are presented in Figures 5 and 6.

We further conduct experiments considering both "gender" and "education" as demographic features, resulting in 4 human decision-maker groups: $S = (0,0)$, $S = (0,1)$, $S = (1,0)$, and $S = (1,1)$. The count of human decision-markers across different groups is shown in Table 4. The utility disparity is measured using the standard deviation of utility distributions across these

Table 4: The details of human-AI interactions dataset (grouped by both "gender" and "education")

| Experiment | Task | Type | Decision Record | Human Decision-makers Count | | | |
|---|---|---|---|---|---|---|---|
| | | | | S=(0,0) | S=(0,1) | S=(1,0) | S=(1,1) |
| 1 | **Art** | image | 4637 | 13 | 64 | 14 | 54 |
| 2 | **Cities** | image | 2878 | 7 | 45 | 6 | 32 |
| 3 | **Sarcasm** | text | 4543 | 13 | 57 | 10 | 62 |
| 4 | **Census** | tabular | 2941 | 10 | 39 | 4 | 39 |

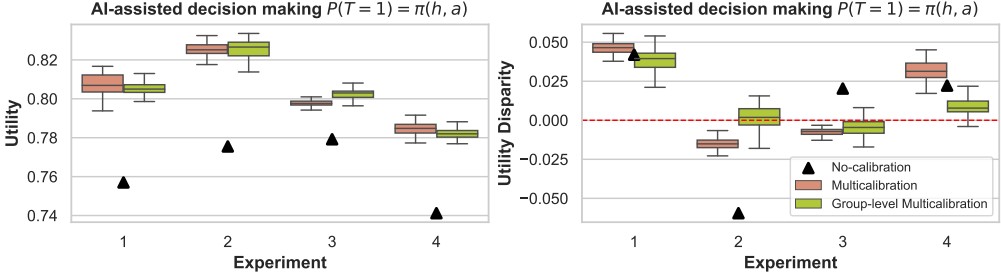

Figure 5: Grouped by "education": Statistics of utility and utility disparity over 100 experiments, where the final decision $P(T = 1) = \pi(h, a)$ is made by human with AI assistance. The AI confidence is either uncalibrated or calibrated using multicalibration and group-level multicalibration, respectively.

subgroups:
$$
\begin{aligned}
\text{Disp} = \text{Std}(&\mathbb{E}[\mathbf{1}(T = Y) \mid S = (0,0)], \mathbb{E}[\mathbf{1}(T = Y) \mid S = (0,1)], \\
&\mathbb{E}[\mathbf{1}(T = Y) \mid S = (1,0)], \mathbb{E}[\mathbf{1}(T = Y) \mid S = (1,1)]).
\end{aligned}
\tag{108}
$$

The results are presented in Figures 7 and 8.

Across these experiments, where human decision-makers are grouped by either single or multiple demographic features, the results align with expectations and demonstrate the following key takeaways:

1. Effectiveness in improving fairness across different settings: Across both single-group and multi-group settings, group-level multicalibration consistently outperforms both uncalibrated and multicalibration methods in terms of fairness. Notably, it avoids the fairness deterioration observed with multicalibration (i.e., tasks 1, 3 and 4 in Figure 5, task 3 in Figure 7).

2. Comparable utility performance: Group-level multicalibration achieves utility performance comparable to multicalibration. For example, in Figure 5, group-level multicalibration demonstrates similar utility performance compared to multicalibration. In tasks 2, and 4 of Figures 7, group-level multicalibration shows superior utility performance. Although there is a slight accuracy drop in task 1, with the 25th percentile decreasing by 0.9% compared to multicalibration, this trade-off is considered acceptable given the significant gains in fairness across groups.

3. Adjusting AI confidence to mitigate utility disparities: Group-level multicalibration effectively adjusts AI confidence to reduce or reverse utility disparities in AI-only decisions, compensating for disparities observed in human-only decisions. This behavior remains consistent across different group settings (Figures 6 and 8).

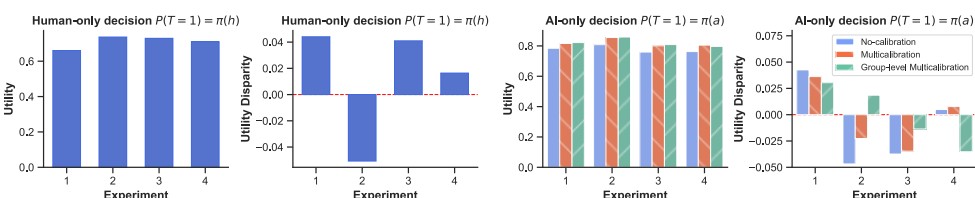

Figure 6: Grouped by "education": The utility and utility disparity where the final decision $P(T=1) = \pi(h)$ is made by human-only or AI-only where AI confidence is either uncalibrated or calibrated using multicalibration and group-level multicalibration, respectively.

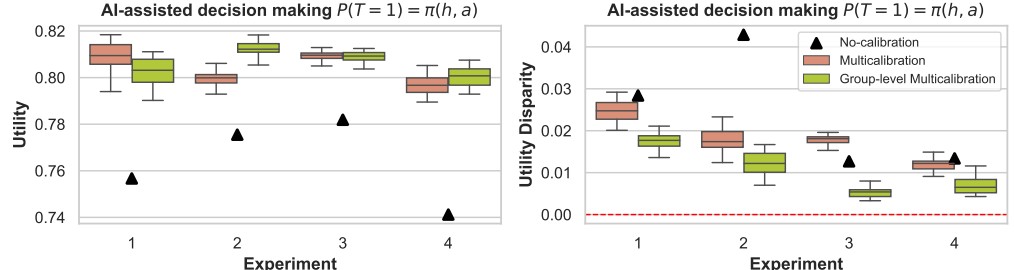

Figure 7: Grouped by both "gender" and "education": Statistics of utility and utility disparity over 100 experiments, where the final decision $P(T=1) = \pi(h,a)$ is made by human with AI assistance. The AI confidence is either uncalibrated or calibrated using multicalibration and group-level multicalibration, respectively.

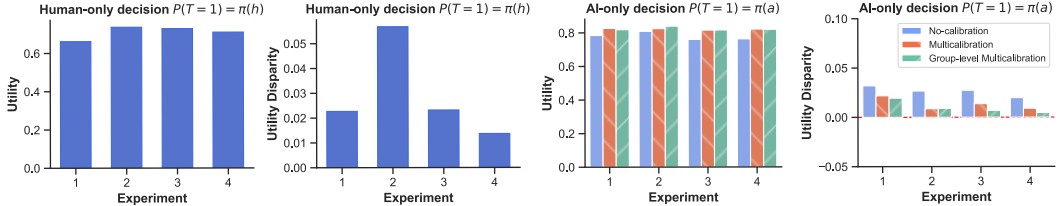

Figure 8: Grouped by both "gender" and "education": The utility and utility disparity where the final decision $P(T=1) = \pi(h)$ is made by human-only or AI-only where AI confidence is either uncalibrated or calibrated using multicalibration and group-level multicalibration, respectively.

