# OpenReview forum: "Human Expertise Really Matters!  Mitigating Unfair Utility Induced by Heterogenous Human Expertise in  AI-assisted Decision-Making"
_ICLR.cc/2025/Conference — ICLR 2025 Conference Withdrawn Submission_

### Official Review · Reviewer_GvKr · 2024-10-28

**Soundness:** 2
**Presentation:** 2
**Contribution:** 2
**Rating:** 5
**Confidence:** 2

**Summary:**

This paper tackles a critical issue in AI-assisted decision-making by exploring the fairness challenges arising from human expertise disparities. The authors introduce a novel approach, group-level multicalibration, aimed at reducing utility disparity across heterogeneous groups of human experts. The concept of inter-group alignment alongside human alignment is proposed to ensure fairer utility distribution, with extensive empirical validation provided across multiple real-world datasets.

**Strengths:**

1. Novel Problem Definition: The paper addresses a unique aspect of AI fairness, focusing on fairness for human experts rather than the subjects of the decisions. This distinction is well-motivated and adds value to AI fairness discussions in decision-support systems.
2. Methodological Contributions: The introduction of inter-group alignment as a complement to human alignment is well-founded theoretically. The group-level multicalibration approach is an innovative addition to the calibration literature, showing promise for balancing utility across user groups with varying expertise.
3. Experimental Rigor: The empirical results demonstrate a significant improvement in utility fairness across groups, with thoughtful evaluation metrics that align well with the theoretical framework.

**Weaknesses:**

The paper is limited to a fairly simple setting, solving a binary decision problem without substantial innovation beyond group-level confidence multicalibration.
1. While the paper presents an interesting problem, it can be challenging to follow, especially in Section 3, which is highly mathematical without a clear exposition of the problem the math addresses. Although this section appears to build on previous research, the authors should clarify these foundations for the reader.
2. The inclusion of $lambda$-discretization may impact the efficiency of group-level multicalibration. Further discussion on its computational implications would be helpful.
3. The authors divided groups into males and females, which they justify. However, from my perspective, the group division could be more nuanced. For example, dividing the population into three or more groups might provide deeper insights, especially if the approach can maintain strong performance despite the additional computational costs.
4. All experiments should be repeated with error bars to convey the variability in results and enhance the robustness of the conclusions.

**Questions:**

1. How was the number of bins chosen in the experiments? Is there a principled approach to this choice?
2. In Figure 2, are there statistically significant differences between the methods, particularly in Experiments 3 and 4?

---

> ### Author Response · Authors · 2024-11-21
> **Author Response**
>
> We appreciate the reviewer’s recognition of the novelty of our problem and the methodological contributions of our work. Thank you for your valuable and thoughtful comments. Please find our point-by-point responses below:
>
> > **Regarding the comment  “The paper is limited to a fairly simple setting”**
>
> We would like to clarify that group-level multicalibration can be applied to non-binary decision settings. The challenge in generalizing from binary decision problems to more complex decision spaces lies in the **identifying appropriate utility function conditions** in such settings (Section 6.2). This requires a social-research into modeling and understanding human utility preferences and behaviors, which is beyond the scope of this work. Therefore, we chose the binary decision setting to **provide a solid foundation** for future exploration of more general decision tasks.
>
> > **Regarding the comment: “Without substantial innovation beyond group-level confidence multicalibration.”**
>
> We respectfully disagree with this point. While group-level multicalibration represents an innovation in our work,  **developing a comprehensive theoretical foundation for identifying and addressing fairness issues represents a more significant and non-trivial contribution**.  We would like to clarify **the detailed novelty** includes:
>
>  1. **The first work focusing on fairness in this context**:  Our work highlights a previously overlooked fairness issue in AI-assisted decision making, which is critical for real-world applications. As discussed in Lines 89–96, ensuring fair utility is essential for advancing AI for social good and improving societal welfare. **Neglecting fair utility can, on one hand, undermine human willingness to adopt AI assistance, and on the other hand, exacerbate utility disparities among decision-makers due to the Matthew Effect, ultimately harming societal welfare.** These factors critically influence the feasibility of AI-assisted decision-making systems in practice, which are unfortunately **ignored by previous work**.
>
> 1. **A novel calibration objective**: We propose a new fairness-driven calibration objective, inter-group alignment, which fundamentally differentiates our theoretical framework and methods from prior work.
>
> 2. **A *tight* theoretical framework of fairness issue in AI-assisted decision making**: We provide **the first in-depth analysis** of the fairness limitations in existing confidence calibration methods. We derive the **tight** upper bound on utility disparity (**which is really non-trivial**), justifying the need of   multicalibration in group-level—**a focus absent in previous studies.**

---

> > ### Author Response · Authors · 2024-11-22
> > **Author Response Continued**
> >
> > >**Regarding the weakness 1 about  "a clear exposition of the problem the math addresses"**
> >
> > Following your suggestion, **we have made the following revisions to the newest submission:**
> > 1. Added subsections in Section 3 to improve structure and make the theoretical content clearer and easier to follow.
> > 2. For definitions based on previous foundations, we have included additional explanations after Definition 3.1 and Definition 3.3 to clarify their mathematical meaning.
> > 3. For our contributions, we have added detailed explanations following Definition 3.5 and Theorem 3.6 to elucidate their purpose and implications better.
> >
> >
> > >**Regarding the weakness 2 about  "discussion on $\lambda$ computational implications"**
> >
> > Thank you for your comment. As mentioned in Lines 327-329, we quantitatively demonstrate the impact of $\widetilde{\alpha}$ and $\lambda$ on the group-level multicalibration level $\alpha$.  In the revised submission, we have also included additional experiments (Appendix A.7) that examine calibration results under varying parameters, including  $\lambda$. These results provide further insights into the computational implications of $\lambda$.
> >
> >
> > > **Regarding the weakness 3 about  "more groups"**
> >
> > We discussed the challenge in the multi-subgroup setting you mentioned in Section 6.2, particularly the uncertainty in defining utility disparity across multiple subgroups. When the form of utility disparity is well-defined, our algorithm can be extended. We selected a specific form of utility disparity—using the standard deviation of utility distributions across subgroups as an example. For this experiment, we considered 4 subgroups defined by both "gender" and "education". **The results, presented in the revised submission (Appendix A.8.2), demonstrate the effectiveness of group-level calibration in multiple subgroups**.
> >
> > > **Regarding the weakness 4 about  "error bars"**
> >
> > Thank you for your suggestion. We would like to clarify that in our experiments, the statistical variability arises from the MLP-fitted decision policy function $\pi$. For the relevant experiments, we conducted $100$ trials with random seeds ($0–99$), as noted in Section 5.2, and **included error bars in the corresponding results (Figure 2).**
> >
> > > **Regarding the question 1 about  "the number of bins"**
> >
> > Thank you for your question. The number of bins in our experiments was chosen based on the following considerations:
> > 1. Avoiding too few bins to ensure finer granularity of the cells, which enhances the effectiveness of group-level multicalibration.
> > 2. Avoiding too many bins to prevent insufficient samples per cell, which could compromise statistical significance.
> > 3. Balancing computational efficiency by preventing excessive binning that could lead to high computational overhead.
> >
> > With these considerations, we set the bin size for human confidence   $h$ to $3$  and for AI confidence $a$ to $1/\lambda$, where $\lambda=0.125$ is a hyperparameter used in the $\lambda$-discretization process. These choices align with the parameter settings in Corvelo, Benz, & Rodriguez (2023). Additionally, as mentioned in our response to Weakness 2, we conducted experiments to examine how different values of  $\lambda$ impact the experimental results, providing further insights into the choice of this hyperparameter.
> >
> > > **Regarding question 2 about  "statistically significant differences in Figure 2"**
> >
> > Yes, there are statistically significant differences observed in Experiments 3 and 4. As illustrated by the boxplots in Figure 2:
> >
> > 1. Utility Disparity: The group-level multicalibration method achieves significantly smaller 25th, 50th (median), and 75th percentiles compared to both the no-calibration and multicalibration methods across 100 experiments, demonstrating improved fairness across human decision-maker groups.
> >
> > 2. Utility: Similarly, group-level multicalibration produces higher 25th, 50th (median), and 75th percentiles compared to the no-calibration and multicalibration methods, indicating improved overall accuracy performance.

---

> > > ### Comment · Reviewer_GvKr · 2024-11-25
> > > **Thank you for detailed response**
> > >
> > > Thank you to the authors for their detailed responses and to the reviewers for their insightful feedback. After reviewing the revised paper and the associated discussions, I remain concerned that the contribution of this work is limited. The experimental results do not demonstrate significant improvement. I suggest that the authors consider conducting statistical tests rather than relying solely on percentile-based analysis, as the observed performance overlaps considerably.
> > >
> > > While the authors introduce heterogeneous expertise group-level multicalibration, this work is restricted to binary decision-making scenarios. Extending this approach to multi-class settings would be far more compelling.

---

> ### Author Response · Authors · 2024-11-25
> **Author Response: More clarifications on your additional suggestions**
>
> Thank you for your feedback. Regarding the two remaining suggestions you still hold, we would like to provide the following clarifications:
>
>
> > **Regarding your concern and suggestion that "The current percentile-based experimental results do not demonstrate significant improvement. I suggest that the authors consider conducting statistical tests rather than relying solely on percentile-based analysis."**
>
> We have **conducted  T-tests** on the outcomes of $100$ repeated experiments (random seeds 0–99) to **assess the statistical significance of differences**. The following table presents the results.
> The **key takeaways** are:
>
>  1）Across all four experiments, our group-level multicalibration consistently demonstrates **higher mean utility** and **lower mean utility disparity**, **outperforming** the baseline multicalibration
>
> 2） The T-tests highlight that **the differences observed in both utility and utility disparity between the two methods are statistically significant with $p-value<0.05$**.
>
> | Experiment | Mean Utility (Multicalibration) | Mean Utility (Group-level Multicalibration (ours)) | t (Utility) | p-value (Utility) | Mean Utility Disparity (Multicalibration) | Mean Utility Disparity (Group-level Multicalibration (ours)) | t (Utility Disparity) | p-value (Utility Disparity) |
> |------------|----------------------------------|----------------------------------------------------|--------------|-------------------|--------------------------------------------|-------------------------------------------------------------|------------------------|-----------------------------|
> | 1          | 0.782                            | **0.790**                                              | -3.095        | **0.006**             | 0.016                                      | **-0.003**                                                     | 7.442                  | **0.000**                       |
> | 2          | 0.817     | **0.840**                                              | -12.053       | **0.000**             | 0.019                                      | **-0.012**                                                     | 7.492                  | **0.000**     |
> | 3          | 0.792    | **0.797**   | -2.908        | **0.009**             | 0.024    | **0.014**               | 3.583                  | **0.002**        |
> | 4          | 0.781                            | **0.784**      | -2.601        | **0.018**             | 0.009                                      | **0.003**                                                      | 3.108                  | **0.006**   |
>
>
>
>
> > **Regarding your concern that our work is "restricted to binary decision-making scenarios" and the suggestion that "extending this approach to multi-class settings would be far more compelling"**
>
>
> We acknowledge that our theoretical results apply only to binary-label decision-making tasks, and extending the analysis of fairness issues in categorical or continuous decision tasks would be an exciting and valuable direction for future work.
> However, as **the first work addressing unfairness arising from human expertise heterogeneity in AI-assisted decision-making**,  we would like to re-clarify **the reasons** behind focusing on binary decision-making scenarios in this work:
>
>
> 1. **Wide applicability of binary decision-making tasks in AI-assisted decision-making**: Binary decision-making tasks are widely used in many real-world applications (e.g., medical diagnostics, loan approval, and hiring decisions). Using these tasks as the focus of our work **provides substantial practical value**.
>
>
>
> 2. **Broadly Acknowledged Challenge in Multi-Class Scenarios**: As clarified in our previous rebuttal, **there is a broadly acknowledged challenge in multi-class settings due to the complexity of identifying appropriate utility function conditions**.   We have included an explicit section, Section 6, describing this challenge. Solid theoretical results depend on robust foundational models.  **Similar challenges** have also been faced in many prior theoretical works [1–5], where **`binary-label tasks are widely adopted  by these previous works [1-5] as a basis for deriving tight theoretical results (which is really non-trivial)`**.
>
> [1] Zhou, Zeyu, et al. "Counterfactual Fairness by Combining Factual and Counterfactual Predictions." **NeurIPS. 2024**.
>
> [2] Donahue, Kate, and Jon Kleinberg. "Optimality and stability in federated learning: A game-theoretic approach." **NeurIPS. 2021**.
>
> [3] Lou, Renze, et al. "Muffin: Curating multi-faceted instructions for improving instruction following." **ICLR. 2023**.
>
> [4] Corvelo Benz, Nina, and Manuel Rodriguez. "Human-aligned calibration for ai-assisted decision making." **NeurIPS. 2023**.
>
> [5] Li, Zhuoyan, Zhuoran Lu, and Ming Yin. "Decoding AI’s Nudge: A Unified Framework to Predict Human Behavior in AI-assisted Decision Making." **AAAI. 2024**.

---

### Official Review · Reviewer_oVxq · 2024-11-02

**Soundness:** 3
**Presentation:** 2
**Contribution:** 3
**Rating:** 6
**Confidence:** 3

**Summary:**

In this work, the authors study a setting in which humans make decisions with the support of algorithmic predictions. However, the authors note that human decision-makers can have varying levels of expertise, which can imply downstream utility disparities when measuring performance w.r.t. the outcomes of decision subjects. The authors propose a multi-calibration framework that mitigates disparities in human decision-makers' utilities by adjusting risk predictions to account for heterogeneity in human decision-making. The authors provide theoretical analysis and experiments which generally justify that the proposed approach mitigates the utility disparity in question.

**Strengths:**

### Motivation.
The authors study a timely and important problem regarding human-algorithm decision-making in scenarios where algorithmic feedback is driving the final human decision. The notion of fairness defined w.r.t. expert decision-maker sub-populations remains under studied, and multi-calibration is a reasonable technical framework to use in this setting.

### Soundness.
The work is theoretically grounded and results are supported by relevant proofs. The work cites relevant related literature, clearly states (most) assumptions, and provides experiments that generally illustrate the utility of the proposed approach.  The experimental evaluation is consistent with the stated aims of the project.

**Weaknesses:**

### Presentation:
- In its current form, the presentation is unnecessarily dense in a way that detracts from the core message of the work. There are a number of formal technical definitions and results that are challenging to digest. For example, Corvelo Benz & Rodriguez, 2023 cover a similar topic in theoretical depth, but the surrounding motivation and discussion is easy to follow.
- I appreciated Figure 1 illustrating the setup with a worked example, but found the current configuration of the figure challenging to parse. I needed to read the rest of the paper before I fully understood the various components and where the numbers in the chart were originating from.
- There are also opportunities to sharpen language and clarify terminology.
    - For example, the draft often uses terms such as “human groups”, “human populations”, and “real-world-users” etcetera without a distinction as to whether the groups in question are the decision-makers (i.e., individuals making selection decisions using algorithmic recommendations) or the decision-subjects (i.e., individuals in the population whom will receive decisions from the decision subjects). Please clarify this in the draft going forward.
    - Line 110: AI confidence facilitates fair utility => is this fair utility w.r.t. the human decision-makers or some other agent (e.g., decision subjects?)
        - E.g., lines 373-377 can be summarized and described in the appendix in more detail if the authors wish.
    - Given the widespread usage of the term alignment in current discourse, I recommend the authors provide a clear and precise definition of alignment early on in the work. The current terms “inter-group alignment” and “”human alignment” (e.g., Fig 1) are not sufficiently specific for me to understand the quantities being modeled.

### Related work:
- This paper generally covers relevant related work. There are several works that study expert heterogeneity in other contexts. For example, Rambachan (2024), Rambachan, Coston, & Kennedy (2024), and Lakkaraju et al. (2018) use heterogeneity as an instrumental variable for identification under unobservables. De-Arteaga et al. (2023) use heterogeneity to mitigate measurement error issues.

- There is an opportunity to sharpen the distinction between this work and Corvelo Benz & Rodriguez (2023). In particular, many results provided by the authors map to those provided by Corvelo Benz & Rodriguez (2023) with an additional conditioning operator on the decision-maker’s demographic subgroup.
- Rambachan 2024, Identifying Prediction Mistakes in Observational Data, https://academic.oup.com/qje/article-abstract/139/3/1665/7682113.
- Rambachan, Coston, & Kennedy 2024, Robust Design and Evaluation of Predictive Algorithms under Unobserved Confounding, https://arxiv.org/abs/2212.09844.
- De-Arteaga et al., Leveraging Expert Consistency to Improve Algorithmic Decision Support, https://arxiv.org/abs/2101.09648.

### Formal setup:
- This framework makes a strong assumption on how humans update their confidence in response to algorithmic confidence scores (i.e., see e.q. 1 in Figure 1). Human decision-making is generally non-parametric and difficult to model. Further, the expertise of human decision-makers can also mediate how they perceive and respond to algorithmic feedback while updating their confidence scores (e.g., Albright 2019). While I appreciate that assumptions are important to drive progress, the assumptions need to be clearly stated and justified.
- The assumption that expert sub-populations depend only upon a single demographic factor severely limits the applicability of the framework. Further, the notion of group-conditional guarantees that hold on a single demographic group is at odds with the traditional multiwcalibration setup, whereby a guarantee holds over many (intersecting) subpopulations.
- How are raw probability values converted to decisions under this framework? Because the utility function is often connected to the choice of threshold, it is important to clarify this relationship and how it impacts the proposed framework. Relatedly, I have concerns regarding how the authors discretize raw scores in the experiments, as this process of discretization implies a utility tradeoff (i.e., false positives to false negatives) that is important to model. Typically a low cutoff implies that false negatives are more costly, while a high cutoff implies that false positives are more costly. See standard cost sensitive classification frameworks for details (e.g., Fernandez et al., 2018).

Fernandez et al. 2018, Cost-Sensitive Learning, https://link.springer.com/chapter/10.1007/978-3-319-98074-4_4
Albright 2019, If You Give a Judge a Risk Score: Evidence from Kentucky Bail Decisions, https://thelittledataset.com/about_files/albright_judge_score.pdf

### Technical Results:
- The current draft presents many technical details without providing a clear narrative motivating their importance and implications. Consider shifting some results to the Appendix and providing more description around the key takeaways. For example, Theorem 3.2 is saying that heterogeneity in human decision-making performance yields disparity in utility. However, I don’t find this result particularly surprising, as this is the relationship that I would expect. I did find Theorem 3.4 more interesting and surprising — i.e., characterizing the specific relationship between notions of alignment and utility. Consider spending more time discussing this result and others in Section 4.
- It would be helpful to clearly state the novelty of these results w.r.t. to Corvelo Benz & Rodriguez, 2023, as they have many similar theoretical results in their work. In particular, how do these results extend our knowledge beyond adding a conditioning operator on Z

### Experiments:
Based on figure three, it is difficult to understand whether there is a meaningful performance improvement under the proposed approach because performance bars do not have confidence intervals and the utility metric is difficult to interpret (i.e., the scaling is quite small for disparity measures).


Rambachan 2024, Identifying Prediction Mistakes in Observational Data, https://academic.oup.com/qje/article-abstract/139/3/1665/7682113.

Rambachan, Coston, & Kennedy 2024, Robust Design and Evaluation of Predictive Algorithms under Unobserved Confounding, https://arxiv.org/abs/2212.09844.

De-Arteaga et al., Leveraging Expert Consistency to Improve Algorithmic Decision Support, https://arxiv.org/abs/2101.09648.

Lakkaraju et al. 2018, The Selective Labels Problem: Evaluating Algorithmic Predictions in the Presence of Unobservables, https://cs.stanford.edu/~jure/pubs/contraction-kdd17.pdf.

**Questions:**

Can you elaborate on how predicted probabilities are thresholded to generate binary decisions under this framework? Further, can you describe the connection between predictive probabilities and utilities under this framework? See comment above for additional context.

Can you elaborate on the specific differences between this work and Corvelo Benz & Rodriguez (2023)?

Can you describe how this framework would be extended to guarantees that hold over multiple demographic groups over experts?

What assumptions does this framework make about how humans incorporate algorithmic feedback into their final decision?

---

> ### Author Response · Authors · 2024-11-21
> **Author Response - Part 1**
>
> We sincerely appreciate your valuable and thoughtful comments. Please find our point-by-point responses below:
>
>
> > **Regrading weakness 1 in the presentation**
>
> Thank you for your helpful suggestions on improving our work presentation. In response, we have made the following revisions in the newest submission:
>
> - Dense Presentation: We have revised Section 3 by adding subsections and restructuring the content to make it clearer and easier to follow.
>
> - Figure 1: We understand your concerns about the difficulty of parsing Figure 1. To address this, we have ensured that the process depicted in the figure is thoroughly described in Lines 78–88.
>
> - Language and Terminology: To sharpen the language and clarify terms, we made the following updates:
>   1. Unified references to "human groups," "human populations," and "real-world users" under the consistent term "human decision-maker groups" for clarity. Thus, "AI confidence facilitates fair utility"  refers to fair utility with respect to human decision-makers.
>   2. Experimental configuration have been moved to Appendix A.7 for further detail.
>   3. Following your suggestion, we replaced "inter-group alignment" and "human alignment" in Figure 1 with the more intuitive phrase "Calibrating AI confidence" to better describe the process represented by the red arrow (calibrating AI confidence to a specified value). Providing a detailed explanation of "inter-group alignment" and "human alignment" in Section 1 might introduce redundancy with the comprehensive discussion in Section 3.  For this reason, we opted to use a simpler and more intuitive term to ensure clarity.
>
> > **Regrading the weakness 2 in related work**
>
> Thank you for your insightful feedback and for pointing out the works by Rambachan et al. and De-Arteaga et al. These studies explore expert heterogeneity in different contexts and provide valuable evidence supporting the existence of heterogeneous experts in practice. They also underscore the importance of addressing the issues raised in our paper. **We have now cited these works in the revised submission (Section 6).**
>
> We also appreciate your suggestion to articulate better the distinction between our work and that of Corvelo Benz & Rodriguez (2023). While their work presents inspiring results on improving utility, **our work addresses a distinct and non-trivial challenge: identifying and addressing fairness issues in AI-assisted decision-making. This focus introduces significant conceptual and methodological differences. Specifically:**
>
> 1. **The first work focusing on fairness in this context**:  Our work highlights a previously overlooked fairness issue in AI-assisted decision making, which is critical for real-world applications. As discussed in Lines 89–96, ensuring fair utility is essential for advancing AI for social good and improving societal welfare. **Neglecting fair utility can, on one hand, undermine human willingness to adopt AI assistance, and on the other hand, exacerbate utility disparities among decision-makers due to the Matthew Effect, ultimately harming societal welfare.** These factors critically influence the feasibility of AI-assisted decision-making systems in practice, which are unfortunately **ignored by previous work**.
>
> 2. **A novel calibration objective**: We propose a new fairness-driven calibration objective, inter-group alignment, which fundamentally differentiates our theoretical framework and methods from prior work.
>
> 3. **A *tight* theoretical framework of fairness issue in AI-assisted decision making**: We provide **the first in-depth analysis** of the fairness limitations in existing confidence calibration methods. We derive the **tight** upper bound on utility disparity (**which is really non-trivial**), justifying the need of   multicalibration in group-level—**a focus absent in previous studies.**
>
> We have revised the submission to clarify these distinctions in Section 6 more explicitly.

---

> ### Author Response · Authors · 2024-11-21
> **Author Response - Part 2**
>
> > **Regrading the weakness 3 in formal setup**
>
> Thank you for your thoughtful comments. We address your concerns point by point below:
> - Assumption on decision policy: we would like to clarity  that the decision policy $P(T=1|h,a)=w\cdot h+(1-w)\cdot a$ in Figure 1 is provided as an intuitive and computationally simple example, not a core dependency of our theoretical framework. To avoid this misunderstanding, we have revised the description in the Lines 84-85.
>
>   Our theoretical framework formally specifies the assumptions on decision policies in Section 2 (Monotone), which accommodates a broad range of decision policies when human decision-makers act rationally. Additionally, in our experiments, we model decision policies using a nonlinear activation function in an MLP, capturing realistic human behavior where the decision policy may be nonlinearly influenced by human confidence and AI confidence.
> - Multi-Subgroup Settings: We discussed the challenge in the multi-subgroup setting you mentioned in Section 6.2, particularly the uncertainty in defining utility disparity across multiple subgroups. When the form of utility disparity is well-defined, our algorithm can be extended. We selected a specific form of utility disparity—using the standard deviation of utility distributions across subgroups as an example. For this experiment, we considered 4 subgroups defined by both "gender" and "education". **The results, presented in the revised submission (Appendix A.8.2), demonstrate the effectiveness of group-level calibration in multiple subgroups**.
> - Conversion of Probabilities to Decisions: We clarify that our framework focuses on a binary decision-making scenario (Line 118). Raw probability values are converted to decisions using the rule:  $T=1$ if $P(T=1)\ge 0.5$, and $T=0$ otherwise. This same conversion is applied in our experiments.
> Based on this conversion, we evaluate utility not by using false positives or false negatives but by using accuracy, $\mathbb{E} \left [\mathbf{1}   (T=Y) \right ]$. Following your suggestion, we have added the description of conversion in Section 2 (Lines 129-131) of the revised submission.
>
> > **Regrading the weakness 4 in technical results**
>
> Thank you for your  suggestions. We address your concerns point by point below:
> - Clear Narrative and Key Takeaways: Following your suggestion, we have revised the manuscript to provide a clearer narrative around the importance and implications of our technical results. Specifically:
>     1. We reduced the emphasis on explaining the fairness limitations of existing calibration methods and added more discussion on the theoretical insights from inter-group alignment and the upper bound on utility disparity.
>     2. Subsections have been added in Section 3 to clarify the connections between different parts and enhance readability.
>
> - Novelty w.r.t. Corvelo Benz & Rodriguez (2023): As mentioned in our response to Weakness 2, our technical results serve for different purpose: fairness, while Corvelo Benz & Rodriguez (2023) focus on utility improvement. We highlights and addresses the fairness limitations of existing calibration methods an propose new  fair calibration objective.
>
>     Our experiments also demonstrate that considering only human alignment  Corvelo Benz & Rodriguez (2023)  may exacerbates fairness issue compared to uncalibrated case (Lines 449–452). This highlights a critical issue for real-world applications and AI for good. We believe our work takes an important step toward advancing fairness in AI-assisted decision-making, a previously unexplored direction.
>
> > **Regrading the weakness 5 in experiments**
>
> We would like to clarify that confidence intervals are not shown in Figure 3 because it focuses on the calibrated  confidence rather than final decision, which remains consistent across different random seeds as it does not involve the variability of final decisions modeled by an MLP (as in Figure 2). The purpose of Figure 3 is to illustrate how group-level multicalibration adjusts the AI confidence compared to traditional multicalibration. Specifically, it demonstrates that group-level multicalibration mitigates utility disparity caused by human-only decisions, either by reducing the disparity or creating an offsetting disparity with the opposite sign.   To further validate this finding, we conducted additional experiments by changing the demographic feature from "gender" to "education." This adjustment causes the results in Figure 3 to change, but the finding remains consistent. These results are included in the revised submission (Appendix A.8.2).
>
> > **Regrading the question 1**
>
> Please refer to the third point in the response to Weakness 3.
>
> > **Regrading the question 2**
>
> Please refer to the response to Weakness 2.
>
> > **Regrading the question 3**
>
> Please see the second point in the response to Weakness 3.
>
> > **Regrading the question 4**
>
> Please see the first point in the response to Weakness 3.

---

> > ### Comment · Reviewer_oVxq · 2024-11-25
> > **Response**
> >
> > Thank you for your detailed response in the rebuttal. While I think there is still a need for improvement in the presentation and clarification of novelty w.r.t. Corvelo Benz & Rodriguez (2023), the revisions make progress in this dimension and this paper does have merit. I have increased my score.

---

### Official Review · Reviewer_Jn4k · 2024-11-03

**Soundness:** 3
**Presentation:** 3
**Contribution:** 3
**Rating:** 6
**Confidence:** 4

**Summary:**

The paper considers the problem of group-wise fairness in human-AI decision-making. The results show that group-wise multi-calibration ensures fairness both theoretically and empirically.

**Strengths:**

The question of fairness in human-AI decision-making is important. The solution is theoretically justified and empirically validated.

**Weaknesses:**

* Necessity of new fairness definitions: Given that the paper achieves fairness via multi-calibration, I'm unsure if the new definitions of intergroup and human alignment are necessary. I suspect these are equivalent conditions to group-wise multi-calibration (see my question below).
* General decision space: while the paper considers binary action space, I think the result might be generalizable to general decision problems if the confidence is interpreted as a probabilistic prediction of the random variable. See
> Ziyang Guo, Yifan Wu, Jason D. Hartline, and Jessica Hullman. 2024. A Decision-Theoretic Framework for Measuring AI Reliance. In Proceedings of the 2024 ACM Conference on Fairness, Accountability, and Transparency (FAccT '24).

* Typo: Thm 3.2 "while AI confidence function f_A is perfect calibration," => is perfectly calibrated.

**Questions:**

* Multi-calibration implies the two fairness conditions in the paper. Are they equivalent?

---

> ### Author Response · Authors · 2024-11-21
> **Author Response**
>
> We thank the reviewer for taking the time and effort to review our paper. Please find below the point-to-point responses to the your comments.
>
>
> > **Regarding weakness 1 and question 1 about "Multi-calibration implies the two fairness conditions in the paper. Are they equivalent?"**
>
> Thank you for your thoughtful comment. We want to clarify the relationship between the fairness conditions and group-level multi-calibration. The two fairness conditions, inter-group alignment, and human alignment serve as the **calibration objectives**, providing the theoretical foundation for fairness. Group-level multi-calibration, on the other hand, acts as **a practical and feasible pathway to achieving these objectives**.
>
> These fairness conditions are **not equivalent** to group-level multi-calibration but rather **justify its effectiveness** in mitigating unfairness in AI-assisted decision-making systems.
>
>
>
> > **Regrading weakness 2 about "generalization to general decision problems if the confidence is interpreted as a probabilistic prediction of the random variable"**
>
> Thank you for your insightful comment. We would like to clarify a misunderstanding: both the human confidence $h$ and AI confidence $a$ in our work are continuous values in the range $[0,1]$ and are random variables interpreted as probabilistic predictions. The binary variable is the final decision $T\in \left \\{   0,1  \right \\} $,
> based on $h$ and  $a$. This corresponds to the "action" or "choice" as described in the work of Ziyang Guo et al. (2024).
> We agree that extending the analysis to utility-fairness issues in categorical or continuous decision tasks would be an exciting and valuable direction for future work.  As discussed in Section 6.2 of our paper, generalizing from binary decision problems to more general decision spaces is challenging, primarily due to the complexity of identifying appropriate utility function conditions in such settings.
>
>
> > **Regrading the weakness 3 about "Typo"**
>
> Thank you for pointing out this typo. We have corrected this in the revised submission.

---

> > ### Comment · Reviewer_Jn4k · 2024-11-21
> >
> > The necessity of fairness definitions is not justified in this paper. The definitions are specifically tuned to a binary decision and binary state setting, and are implied by multi-calibration which is actually used to achieve fairness conditions. In fact, multi-calibration has more guarantee beyond a restricted binary decision problem. If the authors would like to justify such a weaker fairness definition, I'd expect more interesting result about the fairness definitions here beyond implied by multi-calibration, e.g. achieving fairness here is computationally easier than multi-calibration.

---

> ### Author Response · Authors · 2024-11-21
> **Thanks for your feedback**
>
> We would like to clarify that multicalibration is typically used as an **algorithmic fairness** condition to ensure fair predictions across predicted groups, such as ensuring that disease diagnosis results are fair for patients in different racial groups. This notion of fairness is fundamentally different from the fairness we address in this work. Our focus is on the **fairness of utility derived by human decision-makers from AI assistance**. For example,  we focus on **how different groups of doctors can achieve fair diagnostic accuracy when assisted by an AI system, rather than addressing fairness for the patient groups being diagnosed**. Our work aligns more closely with the concept of "allocation fairness" in collaborative settings,  which is **distinct from algorithmic fairness**. Regarding our fairness definition, it is specifically designed to tackle the unique challenges of ensuring fair utility for human decision-makers in AI-assisted setting, an area that remains critical and underexplored.
>
> Furthermore, under the fairness definition proposed in our work—focused on utility fairness for decision-makers—extensive experimental results (Figures 2, 5, and 7) demonstrate that **directly applying multicalibration does not guarantee fairness and may even exacerbate fairness issues**. This is an important point to verify that **our definition of fairness goes beyond the fairness provided by multicalibration**.

---

> ### Comment · Reviewer_Jn4k · 2024-11-21
>
> I totally understand that you care about utility fairness. Blackwell Informativeness Theorem essentially says multicalibration (calibration conditioned on group identity, and, expertise in your case) implies improved utility for all groups. To me, by Blackwell's theorem, achieving the fairness notion for **all decision tasks** with binary states and **not necessarily binary** decisions seems an equivalent condition to multi-calibration (conditioned on group identify and not the one in Benz and Rodriguez), which implies your special case of binary decision. The definitions here are not interesting unless the authors can clarify the statement is not true.

---

> ### Author Response · Authors · 2024-11-22
> **Thanks for your feedback**
>
> Thank you for your constructive comment.
>
> We would like to  `clarify the condition when our fairness notion is not equivalent to multicalibration`: Applying multicalibration in the AI-assisted decision-making context can provide for each human confidence $h\in \mathcal{H}$, the AI confidence $a$ is calibrated to satisfy $\mathbb{E} [P(T=1|h,a)-P(Y=1|h,a)]\le \alpha$ [1].
>
> However, only under a `strong assumption`,
> $P(Y=1|a,h)=P(Y=1|a,h,S=1)=P(Y=1|a,h,S=0)$,
> $\mathbb{E} [P(T=1|h,a)-P(Y=1|h,a)]\le \alpha$ also leads to $\mathbb{E} [P(T=1|h,a,S=1)-P(Y=1|h,a,S=1)]\le \alpha$ and $\mathbb{E} [P(T=1|h,a,S=0)-P(Y=1|h,a,S=0)]\le \alpha$,  and thus becomes equivalent to our fairness notion.
>
> **The issue is that, in practice, the assumption often does not hold.** This issue may not arise in predictors scenarios;  it stems from human behavior, which is **unique to AI-assisted decision-making context**. For example, specific human subgroups may exhibit overconfidence in likelihood estimation, leading to  $P(Y=1|a,h)\not = P(Y=1|a,h,S=1)\not =P(Y=1|a,h,S=0)$.  **Our fairness notion explicitly accounts for such a scenario**.
>
> [1] Hébert-Johnson, Ursula, et al. "Multicalibration: Calibration for the (computationally-identifiable) masses." International Conference on Machine Learning. PMLR, 2018.

---

> > ### Comment · Reviewer_Jn4k · 2024-11-24
> >
> > Thank you for the clarification. I think this assumption needs to be discussed. The following paper shows a similar observation:
> > Pleiss, G., Raghavan, M., Wu, F., Kleinberg, J., & Weinberger, K. Q. (2017). On fairness and calibration. Advances in neural information processing systems, 30.

---

> ### Author Response · Authors · 2024-11-26
> **Author Response**
>
> Thank you for your feedback and for highlighting the [1].
>
> > **Regarding the comment about “I think this assumption needs to be discussed.”**
>
> In Section 2 (Expertise Disparity (ED) and Utility Disparity (UD)), we provide the background of the research problem, focusing on human decision-makers with heterogeneous expertise. **This section captures the assumption discussed in our rebuttal, making our work not equivalent to multicalibration**.
>
> Following your suggestion, **we have added our above discussion in Section 2 to explicitly highlight the unique aspects of our setting**. Please refer to the updated submission for these additions.
>
> > **Regarding the comment about “The following paper shows a similar observation.”**
>
> We would also like to elaborate on the differences between our work and [1]. Beyond the **distinct application contexts**, there are fundamental **differences in problem settings**:
>
> - [1]: The problem setup assumes **two predictors, each trained on data from different groups**.
>
> - Ours: We assume a single AI predictor trained on a dataset containing **data from all groups**.
>
> Given this difference, [1] makes important contributions by investigating the compatibility between **calibration and Equalized Odds (false-positive or false-negative)** under their specific setting. In our work (Lines 107–109),  **equal utility (equal accuracy)** involves simultaneously constrains **human-alignment and inter-group alignment** below a pre-specified threshold. Our theoretical results show that group-level multicalibration provides a practical way to compatibly achieve both human-alignment and inter-group alignment.
>
> These discussions are now **summarized in Related Work (Section 6) of the revised submission**.
>
> [1] Pleiss, G., Raghavan, M., Wu, F., Kleinberg, J., & Weinberger, K. Q. (2017). On fairness and calibration. Advances in neural information processing systems, 30.

---

### Official Review · Reviewer_fiVs · 2024-11-04

**Soundness:** 3
**Presentation:** 2
**Contribution:** 2
**Rating:** 6
**Confidence:** 3

**Summary:**

The work investigates the role of calibration in AI-assisted decision-making, focusing on how the alignment between AI confidence scores and accuracy impacts human decisions when human decisions vary across groups. The authors analytically show that, when different groups of users have different decision policies, using the same AI confidence alignment for both groups cannot guarantee the same "utility" across groups. They also empirically show that multicalibration of AI confidence with respect to each group improves overall calibration of the decision policy.

**Strengths:**

The scope of the paper is clear and the paper itself tackles an important and timely problem, definitely relevant to the ICLR community. The paper also goes deep into the details about why multicalibration --as done by previous work-- can fail. The paper also dives into the details through both analytical and empirical analyses, and the results were interesting (although more explanations about them would be helpful).

**Weaknesses:**

Novelty: The novelty of the paper is one of the main weaknesses. The authors propose to multicalibrate the AI confidence separately for each subgroup of users; if my understanding is correct, this could already be done by previous work just by considering the subgroup information as a feature. This is done in standard multicalibration in tabular data problems. In addition, the paper is very similar to the recent work of Benz and Rodriguez (2023). Algorithm, experiments, etc look similar. The main novelty in this work is that the authors show that the AI confidence function will not be calibrated when decision policies vary across groups and thus recommend running the multicalibration algorithm by group. Is this correct?

Clarity: Presentation can be improved. For example, in section 3 multiple definitions and theorems are presented without much descriptions or examples. In many parts, the paper is very dense and notation is heavy. The expectation is never defined, as well as other mathematical terms are not clearly explained. Some examples would help the reader. What is the explanation behind the upper bound in Theorem 3.6? Can the authors provide an intuition behind this result?

Typos: Lines 126-127, the policy is not binary. Line 192: calibration -> calibrated. There are others around the manuscript.

I recommend adding some explanations in the code to help the reviewer or the interested reader understand what’s happening. As a reviewer, I found it really hard to navigate it.

**Questions:**

Questions are listed above. One additional question: The authors focus on the case where each individual belongs to one subgroup. In practice, even in the case of demographic subgroups, each individual belongs to many subgroups (eg gender, race, etc). Multicalibration handles these scenarios, but I do not get a sense of whether the authors’ algorithm extends to this case. Can the authors discuss this point?

---

> ### Author Response · Authors · 2024-11-21
> **Author Response - Part 1**
>
> We thank the reviewer for finding our work essential and timely and for finding the results interesting. We also appreciate the detailed comments posed by the reviewer. Please find below the point-to-point responses to your comments.
>
> **Weaknesses 1 (Clarifying the novelty)**
> > **Regarding the comment: “The authors propose to multicalibrate the AI confidence separately for each subgroup of users; if my understanding is correct, this could already be done by previous work just by considering the subgroup information as a feature. This is done in standard multicalibration in tabular data problems.”**
>
> We appreciate the reviewer’s comment and would like to clarify a potential misunderstanding. Prior works such as multicalibration **do not**  incorporate subgroup information into the calibration objective, overlooking the critical necessity of considering subgroup-specific nuances to achieve fairness.
>
> Our findings reveal that applying multicalibration alone cannot guarantee fairness (Theorem 3.4). In fact, as demonstrated in Section 5.2, it can sometimes exacerbate fairness issues due to the lack of explicit subgroup handling.
>
> To address these limitations, one of our contributions is to introduce group-level multi calibration. While this approach is conceptually straightforward and builds upon multicalibration, we demonstrate theoretically and empirically **why multicalibration in group-level  is not just feasible but necessary** for effectively mitigating unfairness.
>
>
> > **Regarding the comment: "The main novelty in this work is that the authors show that the AI confidence function will not be calibrated when decision policies vary across groups."**
>
> We respectfully disagree with this point.  While  "the AI confidence function may not be calibrated when decision policies vary across groups" is a new finding of our work, **developing a comprehensive theoretical foundation for identifying and addressing fairness issues represents a more significant and non-trivial contribution**.  We would like to clarify **the detailed novelty** distinct our work from previous  include:
>
>  1. **The first work focusing on fairness in this context**:  Our work highlights a previously overlooked fairness issue in AI-assisted decision making, which is critical for real-world applications. As discussed in Lines 89–96, ensuring fair utility is essential for advancing AI for social good and improving societal welfare. **Neglecting fair utility can, on one hand, undermine human willingness to adopt AI assistance, and on the other hand, exacerbate utility disparities among decision-makers due to the Matthew Effect, ultimately harming societal welfare.** These factors critically influence the feasibility of AI-assisted decision-making systems in practice, which are unfortunately **ignored by previous work**.
>
> 2. **A novel calibration objective**: We propose a new fairness-driven calibration objective, inter-group alignment, which fundamentally differentiates our theoretical framework and methods from prior work.
>
> 3. **A *tight* theoretical framework of fairness issue in AI-assisted decision making**: We provide **the first in-depth analysis** of the fairness limitations in existing confidence calibration methods. We derive the **tight** upper bound on utility disparity (**which is really non-trivial**), justifying the need of   multicalibration in group-level—**a focus absent in previous studies.**
>
> We have **revised** the submission to clarify the distinctions more explicitly in Section 6.
>
> > **Regarding the comment, "In addition, the paper is very similar to the recent work of Benz and Rodriguez (2023). Algorithm, experiments, etc look similar."**
>
> As outlined above, one of our key innovations is developing a comprehensive theoretical foundation that justifies the necessity of multicalibration at the group level. Based on this, we propose **group-level multicalibration**. While conceptually straightforward and building upon the framework of multicalibration (Hebert-Johnson et al., 2018), our approach is specifically designed to **provide a practical pathway to achieving the fair calibration objectives outlined in Section 3**.
>
> **In contrast**, Benz and Rodriguez (2023) utilized the **original multicalibration** method to achieve a **utility-optimal calibration objective**. Apparently, our work pursues distinct calibration objectives. We explicitly highlight the distinctions in the revised submission (Lines 299-304).
>
> Regarding the experiments, we refer to Benz and Rodriguez (2023) as a baseline and follow their experimental setup (as explicitly cited in Lines 375,406,1220). While Benz and Rodriguez (2023) primarily evaluate aggregate utility, we focus is on utility-fairness across subgroups rather than aggregate utility alone. The experimental results from their multicalibration method serve as our baseline, and our results clearly demonstrate significant advantages in fairness over multicalibration.

---

> ### Author Response · Authors · 2024-11-21
> **Author Response - Part 2**
>
> > **W2 (Clarifying the presentation)**
>
> Thank you for your valuable suggestions regarding clarity and presentation. Following your comment, we have made **the following revisions** to improve the paper's readability and address the specific concerns:
>
> - Added subsections in Section 3 to make the theoretical content clearer and easier to follow.
> - Provided additional explanations for Definition 3.5 in Section 3 to help clarify its role and importance.
> - Expanded the explanations for Theorem 3.6 in Section 3, including an intuition behind the upper bound to enhance understanding.
> - Defined key mathematical terms in Section 2---Human-AI interactive model in AI-assisted decision-making.
>
> Please review our revised submission.
>
>
>
> > **W3 (Clarifying the typos)**
>
> Thank you for your comment. We want to clarify the misunderstanding regarding Lines 130–131 (previously referenced as Lines 126–127). Here, we refer to the decision variable $T$, which is binary, taking values of either 0 or 1, rather than the policy $\pi$.
> Additionally, thank you for pointing out the typo in Line 189 (previously referenced as Line 192). We have corrected this to replace "calibration" with "calibrated."
>
> > **W4 (Adding explanations in the code)**
>
> We have included additional explanations within the code to enhance its readability. For further details, please refer to the anonymous link in the paper.
>
> > **Q1 (Extending to multi-subgroups)**
>
> Thank you for your question. We discussed the challenge in the multi-subgroup setting you mentioned in Section 6.2, particularly the uncertainty in defining utility disparity across multiple subgroups. When the form of utility disparity is well-defined, our algorithm can be extended. We selected a specific form of utility disparity—using the standard deviation of utility distributions across subgroups as an example. For this experiment, we considered four subgroups defined by "gender" and "education". **The results, presented in the revised submission (Appendix A.8.2), demonstrate the effectiveness of group-level calibration in multiple subgroups**.

---

> > ### Comment · Reviewer_fiVs · 2024-11-21
> >
> > Thanks for the thoughtful response. My main concerns have been addressed and I have raised my score.

---

### Author Response · Authors · 2024-11-21
**General Response: Novelties and Generalization to Multiple Groups**

Dear Chairs and Reviewers,

We sincerely thank all the reviewers for their time and valuable feedback on our work.

We would like to **address a common misunderstanding regarding group-level multicalibration as the main contribution of our work and concerns about its novelty.** Inspired by multicalibration, we propose the group-level multicalibration method in Section 4 as a practical pathway to achieving the fair calibration objectives outlined in Section 3. **While group-level multicalibration demonstrates promising performance and is one of the contributions of our work, developing a comprehensive theoretical foundation for identifying and addressing fairness issues represents a more significant and non-trivial contribution.** The **key novelties** distinguishing our work from previous research include:

1. **The first work focusing on fairness in this context**:  Our work highlights a previously overlooked fairness issue in AI-assisted decision making, which is critical for real-world applications. As discussed in Lines 89–96, ensuring fair utility is essential for advancing AI for social good and improving societal welfare. **Neglecting fair utility can, on one hand, undermine human willingness to adopt AI assistance, and on the other hand, exacerbate utility disparities among decision-makers due to the Matthew Effect, ultimately harming societal welfare.** These factors critically influence the feasibility of AI-assisted decision-making systems in practice, which are unfortunately **ignored by previous work**.

2. **A novel calibration objective**: We propose a new fairness-driven calibration objective, inter-group alignment, which fundamentally differentiates our theoretical framework and methods from prior work.

3. **A *tight* theoretical framework of fairness issue in AI-assisted decision making**: We provide **the first in-depth analysis** of the fairness limitations in existing confidence calibration methods. We derive the **tight** upper bound on utility disparity (**which is really non-trivial**), justifying the need of   multicalibration in group-level—**a focus absent in previous studies.**

Another common concern raised by reviewers relates to applying our approach to multiple human decision-maker groups.   **We included some additional results in the revised submission that move things in a positive direction in this regard**. Please refer to Appendix A.8 for further details.

**Other valuable comments from the reviewers are responded to point-by-point below.**

Below, we summarize **`a list of revisions made in the newest updated submission`** for your review.

> **Overall**
> Unified references to "human groups," "human populations," and "real-world users" under the consistent term "human decision-maker groups" for clarity.

> **Section 1**
> - **Lines 84–85**: Clarified that the decision policy in Figure 1 is provided as an illustrative example and is not a core dependency of the theoretical framework.
> - **Figure 1**: Replaced "Inter-group alignment + human alignment" with the more intuitive phrase "Calibrating AI confidence" for better clarity.

> **Section 2**
> - **Lines 129–131**: Clarified the process of converting raw probabilities into binary decision.

> **Section 3**
> - **Overall**: Added subsection titles to improve structure and make the theoretical content clearer and easier to follow.
> - **Definition 3.1 and Definition 3.3**: Included additional explanations to clarify their mathematical meaning.
> - **Theorem 3.2**: Modified the mathematical expression to a more concise form.
> - **Definition 3.5 and Theorem 3.6**: Added detailed explanations to elucidate their purpose and implications.
> - **Line 189**: Corrected "calibration" to "calibrated."

> **Section 4**
> - **Lines 299–304**: Explicitly highlighted the distinctions between group-level multicalibration and multicalibration.

> **Section 5**
> - **Line 351-353**: Added hyperparameter configuration.
> - **Line 449-452**: Explicitly demonstrated multicalibration fails to provide fairness based on experimental observations.

> **Section 6**
> - **Line 484-494**: Cited additional related works (Rambachan et al. and De-Arteaga et al.) and clarified the distinctions between our work and Corvelo Benz & Rodriguez (2023).

> **Appendix**
> - **Appendix A.7 (Experiment Settings)**: Moved experimental setup details previously in Section 5 to the newly added Appendix A.7 for better organization.
> - **Appendix A.8 (Additional Experiments)**: Added new experiments to validate the **hyperparameter impact** and **generalization** of group-level multicalibration.

Thank you again for your constructive comments.

Kind regards,

The authors

---

### Note · Authors · 2025-01-22

I have read and agree with the venue's withdrawal policy on behalf of myself and my co-authors.